# Primary Arterial Hypertension and Drug-Induced Hypertension in Philadelphia-Negative Classical Myeloproliferative Neoplasms: A Systematic Review

**DOI:** 10.3390/biomedicines11020388

**Published:** 2023-01-28

**Authors:** Mihnea-Alexandru Găman, Vincent Kipkorir, Bahadar S. Srichawla, Arkadeep Dhali, Amelia Maria Găman, Camelia Cristina Diaconu

**Affiliations:** 1Faculty of Medicine, “Carol Davila” University of Medicine and Pharmacy, 050474 Bucharest, Romania; 2Department of Hematology, Center of Hematology and Bone Marrow Transplantation, Fundeni Clinical Institute, 022328 Bucharest, Romania; 3Department of Human Anatomy and Physiology, University of Nairobi, Nairobi 30197, Kenya; 4Department of Neurology, UMass Chan Medical School, Worcester, MA 01655, USA; 5Department of Internal Medicine, Nottingham University Hospitals NHS Trust, Nottingham NG7 2UH, UK; 6Department of Pathophysiology, University of Medicine and Pharmacy of Craiova, 200349 Craiova, Romania; 7Clinic of Hematology, Filantropia City Hospital, 200143 Craiova, Romania; 8Department of Internal Medicine, Clinical Emergency Hospital of Bucharest, 105402 Bucharest, Romania

**Keywords:** myeloproliferative neoplasms, polycythemia vera, essential thrombocythemia, primary myelofibrosis, hypertension, blood pressure, cardiovascular risk factors, thrombosis, antihypertensive agents, drug-induced hypertension

## Abstract

The impact of primary arterial hypertension (HTN) in myeloproliferative neoplasms (MPNs) remains unclear, with scant literature available, mostly focusing on cardiovascular risk factors as a singular entity or on organ-specific HTN. Furthermore, available studies reporting findings on drug-induced HTN in MPNs report varying and contradictory findings. In consideration of the above, this study set out to systematically review the available literature and shed light on the occurrence of HTN in MPNs, its association with thrombosis, as well as the drugs used in MPN management that could increase blood pressure. The literature search yielded 598 potentially relevant records of which 315 remained after the duplicates (*n* = 283) were removed. After we screened the titles and the abstracts of these publications, we removed irrelevant papers (*n* = 228) and evaluated the full texts of 87 papers. Furthermore, 13 records did not meet the inclusion criteria and were excluded from the systematic review. Finally, a total of 74 manuscripts were entered into the qualitative synthesis and included in the present systematic review. Our systematic review highlights that HTN is the most common comorbidity encountered in MPNs, with an impact on both the occurrence of thrombosis and survival. Moreover, drug-induced HTN remains a challenge in the management of MPNs. Further research should investigate the characteristics of patients with MPNs and HTN, as well as clarify the contribution of HTN to the development of thrombotic complications, survival and management in MPNs. In addition, the relationship between clonal hematopoiesis of indeterminate potential, HTN, cardiovascular disease and MPNs requires examination in upcoming assessments.

## 1. Introduction

Philadelphia-negative myeloproliferative neoplasms (MPNs) refer to a set of acquired hematological clonal disorders where the presence of abnormal hematopoietic stem cells leads to the malignant transformation of myeloid progenitors and subsequent increased production of myeloid cells [1]. Their growing global burden, especially for polycythemia vera (PV), essential thrombocythemia (ET) and primary myelofibrosis (PMF), continues to be a concern. More importantly, the impact of comorbidities in such patients plays a significant negative effect on prognosis and overall survival rates [2]. Of these, cardiovascular comorbidities such as hypertension (HTN) with concomitant cerebrovascular events remain the most common and are attributed to about 45% of deaths in MPN patients [3]. The association between MPNs and HTN stems from a myriad of factors, both known and unknown. Genetically, the *JAK2V617F* mutation seen in most MPNs has been implicated, with resultant vascular neutrophil accumulation and elastic lamina degradation [4]. Furthermore, the known associations between MPNs and an altered cardiac physiology as well as an impaired neural control of blood pressure (BP) play a key role in this phenomenon [5]. Another factor to be considered is the characteristically high platelet count and platelet dysfunction in MPNs, which predisposes patients to thrombosis and subsequent development of HTN [6,7]. In addition, an overlooked cause of HTN in MPNs includes the drugs used in the treatment of these conditions. By inhibiting the JAK2/STAT3 pathways, responsible for smooth and cardiac muscle as well as endothelial functioning, drugs used in the management of MPNs alter the expression of endothelial nitric oxide synthase (NOS), hence aggravating BP [8,9]. Owing to the above, MPN patients with HTN develop an altered neurohormonal axis, characterized by diminished sympathetic nervous system activity, as a reflex to counter HTN that co-exists with MPNs [10,11]. The consequences of HTN and thrombosis in MPNs cannot be understated. Their associated end-organ damage and dysfunction is often accompanied by significant impairments in prognosis and survival rates. Adverse outcomes of the aforementioned, e.g., glomerulopathy, arterial damage and occlusion as well as kidney failure, have been widely observed [12,13].

Considering that MPNs are generally blood cancers encountered in the elderly population who also displays numerous cardiovascular risk factors (CVRFs), e.g., HTN, type 2 diabetes mellitus (T2DM), dyslipidemia (in particular hypercholesterolemia), smoking and others, and subsequent cardiovascular disease, understanding the epidemiology and the impact of each CVRF on the risk of thrombosis and on survival requires further investigation [14,15]. A current caveat of the research focused on the assessment of CVRFs in MPNs is that most investigations published so far have explored the contribution on thrombotic events and risk of death of CVRFs as a singular entity, although their epidemiology in MPNs is probably distinct, similar at least to the prevalence of CVRFs in the adult population. As HTN is the most common CVRF encountered in the general adult population [16], we believe that analyzing its epidemiology, as well as contribution to thrombosis occurrence and impact on survival, should emerge as the first step in tracking down the relevance of each CVRF in MPNs. In addition, physicians should also be aware of drug-induced HTN which can develop as a side effect of the medications currently employed to treat individuals suffering from MPNs. Thus, the impact of primary arterial HTN in MPNs, however, still remains unclear, with scant literature available, mostly focusing on CVRFs as a singular entity or on organ-specific HTN. Furthermore, available studies reporting findings on drug-induced HTN in MPNs have also depicted varying and contradictory findings. In consideration of the above, this study set out to systematically review the available literature and shed light on the occurrence of HTN in MPNs and its association with thrombosis, as well as the impact on BP of the drugs used in MPN management. To our view, one of the main novelties of this manuscript is that it is the first systematic review to provide a comprehensive overview of primary arterial HTN and drug-induced HTN in MPNs.

## 2. Materials and Methods

In order to conduct this systematic review of the literature, we followed the recommendations set forth in the Preferred Reporting Items for Systematic reviews and Meta-Analyses (PRISMA) guidelines [17]. The protocol of this study was registered in PROSPERO (registration number: 333920).

Three assessors independently searched the PubMed/MEDLINE, Web of Science and SCOPUS databases to identify relevant articles published from the inception of these databases up to 1 May 2022. The search strategy was based on a selection of relevant keywords and word combinations. In addition, the reference lists of the previously identified publications or of relevant reviews were hand-searched in order to track down any potentially suitable manuscripts that could enrich the content of this systematic review.

We included publications in the qualitative synthesis if: 1. They were original articles or research letters that provided information on the epidemiology of HTN in MPNs, the impact of HTN on the occurrence of thrombotic events in MPNs or discussed the management of HTN in MPNs. Case reports were generally excluded and only included in the analysis if they presented valuable information not specified elsewhere, e.g., the management of HTN emergencies in MPNs; 2. The subjects were diagnosed with Philadelphia-negative classical MPNs: PV, ET, PMF or secondary myelofibrosis (SMF), i.e., post-PV myelofibrosis (PV-MF) or post-ET myelofibrosis (ET-MF); 3. The patients were diagnosed with HTN (primary or drug-induced); 4. The subjects were adults (aged ≥ 18 years); 5. The papers were published in English, Romanian, French or Italian (the languages spoken by the authors); 6. The full text of the publications could be downloaded or retrieved. We opted for the next exclusion criteria: 1. Reviews, case reports, meeting abstracts, grey literature; 2. Non-clinical investigations (in vitro or animal studies). 3. Studies conducted in children (age < 18 years); 3. The papers were published in languages unknown to the authors (e.g., Chinese, Czech, Hungarian); 4. The papers did not report sufficient data on the outcomes of interest (HTN in MPNs, association with thrombosis, HTN treatment options); 5. The papers examined causes of HTN (secondary HTN, with the exception of drug-induced HTN which was tackled in this systematic review); 6. The full texts of the papers could not be retrieved.

We assessed the eligibility of the identified publications by screening their titles and abstracts. After this stage, the full texts of the eligible papers were retrieved and examined by three assessors who additionally completed a customized Microsoft Office Excel 2013 spreadsheet by extracting the next information from the eligible manuscripts: first author surname, publication year, study location, study design, type of MPN, number of MPN patients recruited, prevalence of HTN, occurrence of thrombotic events, main results of the analyzed papers. Any disagreement was resolved by consensus with a fourth investigator.

We assessed the methodological quality of the papers using the methodological index for non-randomized observational studies (MINORS) [18] and the risk of bias using the Mixed Methods Appraisal Tool (MMAT) [19].

The heterogeneity of the included manuscripts in terms of aim, study design and outcome measures prohibited a meta-analysis. However, we systematically reviewed the eligible papers and presented them under different categories as detailed in the Results section.

## 3. Results

The literature search yielded 598 potentially relevant records of which 315 remained after duplicates (*n* = 283) were removed. After we screened the titles and the abstracts of these publications, we removed irrelevant papers (*n* = 228) and evaluated the full texts of 87 papers. Furthermore, 13 records did not meet the inclusion criteria and were excluded from the systematic review. Finally, a total of 74 manuscripts were entered into the qualitative synthesis and included in the present systematic review [5,10,11,13,20,21,22,23,24,25,26,27,28,29,30,31,32,33,34,35,36,37,38,39,40,41,42,43,44,45,46,47,48,49,50,51,52,53,54,55,56,57,58,59,60,61,62,63,64,65,66,67,68,69,70,71,72,73,74,75,76,77,78,79,80,81,82,83,84,85,86,87,88,89]. A flowchart diagram of the detailed steps of the literature search process is depicted in Figure 1.

### 3.1. Hypertension: Just a Comorbidity in Myeloproliferative Neoplasms?

In terms of HTN epidemiology in MPNs, it seems that elevated BP is one of the most frequent comorbidities encountered in this category of blood cancer patients. Mitra et al. reported that more than half of individuals (52% in those with splenomegaly and 50% in those without) diagnosed with PMF/SMF display elevated BP, with HTN ranking as the most common comorbidity in these patients [20]. Similarly, HTN was the most prevalent comorbidity in Parasuraman et al.’s research who evaluated 7718 patients diagnosed with PV from the United States Veterans Health Administration database and recorded the presence of HTN in ~72% (*n* = 5531) of the study group [21]. Similarly, the investigators of the Prospective Observational Study of Patients With Polycythemia Vera in US Clinical Practices (REVEAL) who recruited 2510 individuals suffering from PV reported HTN as the most prevalent comorbidity in their cohort, affecting ~71% (*n* = 1772) of the participants [22]. Data derived from another United States database of PV subjects (*n* = 2856) of whom ~57 (*n* = 1630) were diagnosed with elevated BP revealed that HTN was more prevalent in high-risk vs. low-risk PV (~65% vs. ~43%, *p* < 0.001) [23]. Likewise, Accurso et al. retrospectively evaluated a cohort of 603 MPN subjects hospitalized in a hematology reference center from Italy (University Hospital Policlinico “Paolo Giaccone”, Palermo, Italy) and concluded that HTN was indeed the most frequently encountered CVRF in MPNs, being present in ~63% of PV, ~64% of ET, ~15% of pre-fibrotic PMF and ~63% of PMF/SMF cases [24]. In addition, the investigators of the Myelofibrosis and Essential Thrombocythemia Observational STudy (MOST) reported that 56% of the 1207 ET individuals recruited from United States-based hospitals suffer from HTN which, thus, leads the ranking of comorbid conditions in ET [25].

HTN ranked as the most common comorbidity in PV subjects in other parts of the world, such as India (*n* = 24/52 patients, 46%) [26]. Data from other Asian countries, e.g., Malaysia, revealed that there is an association of HTN with the presence of the *JAK2V617F* mutation (OR = 1.688; 95% CI = 1.234–2.310; *p* = 0.001) and particularly with the diagnosis of PV and ET. HTN was the most common comorbidity among MPN subjects, i.e., PV > ET > PMF > MPN unclassifiable (MPNu) > hypereosinophilic syndrome (~58% vs. ~42% vs. ~41% vs. ~36% vs. ~33%, *p* < 0.001) [27]. In Pakistan, ~90% of the individuals diagnosed with PV suffer from HTN which was associated with the age of the patients (*p* = 0.033) [28]. Similarly, HTN was the most common CVRF in Mancuso’s cohort of 233 individuals diagnosed with ET, affecting ~40% of the research sample. Moreover, in ~40% of the ET group, the investigators identified the presence of one CVRF, whereas ~35% had multiple CVRFs. The authors also compared several of the available instruments for the prediction of thrombotic complications in ET, namely the traditional prediction score which takes into account the age of the patient and the history of thrombosis, the International Prognostic Score for Essential Thrombocythemia thrombosis (IPSET-thrombosis) score, that evaluated not only the age and history of thrombosis, but also the presence of the *JAK2V617F* mutation and CVRF, as well as the revised IPSET-thrombosis score, that assessed the age, the history of thrombosis and the presence of the *JAK2* wild-type or V617F mutation. When the IPSET-thrombosis score was calculated, several subjects who were assigned to the group at low-risk of thrombotic events were reclassified as intermediate or high-risk which is in line with the conclusion of the researchers who pointed out that thrombotic complications occurred mainly in ET with associated CVRFs [29].

HTN was also the most common comorbidity (56%) in patients diagnosed with myeloid cancers (50% of the study group consisted of MPN and MPN/myelodysplastic sydrome cases) and who developed Coronavirus Disease 2019 (COVID-19), but its presence had no impact on survival [30]. Likewise, in a British cohort of COVID-19 patients with malignant (including MPNs) and benign blood disorders of whom 55% died, although HTN was the first-ranked comorbidity (41%), its presence was not linked with an adverse outcome [31]. Moreover, although HTN seems to be the most common comorbidity in MPNs (57.3% of 192 PV and ET patients), it is more likely to be present in PV and ET individuals who are currently receiving statins as lipid-lowering drugs (~85% vs. ~51%, *p* < 0.001), probably due to the increased burden of CVRFs in MPNs who associate dyslipidemia [32]. However, Palandri et al. had previously reported that HTN, which occurred in 48% of the 386 investigated individuals with ET, independently predicted poor survival [33].

Wojcicki et al. and Lewandowski evaluated the activity in the sympathetic nervous system and the perfusion of the retina and kidneys in patients diagnosed with PV who also suffered from HTN versus a control group of HTN patients without PV but matched for sex, age, BP values and number of medicines used. Three quarters of the recruited PV subjects exhibited high BP values when examined in the investigator’s offices or based on ambulatory BP measurements but had a less notable BP decrease during the night, as well as lower heart rate values per 24 h (the heart rate during the day was particularly lower in PV) and a reduced microneurography-evaluated muscle sympathetic nervous activity (the muscle activity per minute was particularly lower in PV and was positively associated with the heart rate during the night, r = 0.617, *p* < 0.05) when compared to the controls. In addition, the authors detected lower serum-free epinephrine and aldosterone concentrations (*p* = 0.048) versus HTN non-PV patients. Although there were no differences regarding the morphology of the retina between the two sample groups, the PV subjects exhibited a decreased retinal microperfusion (*p* = 0.032) which was negatively associated with the number of erythrocytes in the PV group (r = −0.44, *p* = 0.012). The resistive index of the kidney arteries was also elevated in the PV versus non-PV HTN counterparts (*p* = 0.033) [10,11]. In addition, the presence of HTN may also aggravate MPN-related glomerulopathy via nephrosclerosis with the development of polar vasculosis and exudative lesions [34]. Although Patino-Alonso reported in a propensity-analysis similar values for SBP and DBP between MPN subjects (*n* = 57) and healthy counterparts (*n* = 114), they discovered that in MPNs there is an elevated risk of carotid artery injury (OR = 2.382, 1.066–5.323, *p* = 0.034) and an elevated albumin-to-creatinine ratio as compared to the control group, signaling that in these blood cancers there is a need to screen for target organ damage [35]. Moreover, the association of MPNs and HTN can also contribute to the occlusion of the central artery of the retina and present as a sudden loss of vision [36]. Kidney dysfunction is also a common encounter in MPNs and is directly related to the presence of HTN. Data from the German Study Group for MPN Bioregistry which included 1420 individuals of which 49% had HTN, pointed out that elevated BPs were more common in PV vs. ET or vs. PMF/SMF (~63% vs. ~46% and ~63% vs. ~39%, respectively, *p* < 0.0001). Most MPN subjects had eGFR = 60–89 mL/min/1.73 m^2^ (~57%), followed by eGFR < 60 mL/min/1.73 m^2^ (~22%) and eGFR ≥ 90 mL/min/1.73 m^2^ (~21%). More PMF/SMF subjects had eGFR < 60 mL/min/1.73 m^2^ vs. PV or vs. ET (~28% vs. ~20%, *p* = 0.005 for both). Overall, HTN was a risk factor (OR = 2.419, 95% CI 1.879–3.114, *p* < 0.0001) for kidney dysfunction, particularly in subjects with eGFR < 60 mL/min/1.73 m^2^ vs. eGFR ≥ 90 mL/min/1.73 m^2^ (OR = 6.220, 95% CI 1.501–25.779, *p* = 0.01) and vs. eGFR = 60–89 mL/min/1.73 m^2^ (OR = 3.429, 95% CI 1.207–9.739, *p* = 0.02) in the univariate regression analysis. Following multiple regression, HTN persisted as a risk factor for kidney dysfunction (OR = 2.004, 95% CI 1.440–2.789, *p* < 0.0001). HTN emerged as a risk factor for thrombotic (OR = 1.838, 95% CI 1.398–2.418, *p* < 0.0001 in the univariate and OR = 1.800, 95% CI 1.349–2.401, *p* = 0.01 in the multivariate regression) but not for bleeding complications (OR = 1.302, 95% CI 0.678–2.499, *p* > 0.05). In particular, HTN was a predictor for thrombotic events in ET (OR = 1.913, 95% CI 1.099–3.330, *p* = 0.02) and PMF/SMF (OR = 1.960, 95% CI 1.067–3.601, *p* = 0.03) [12].

Kwiatkowski et al. also revealed that ET subjects display a degree of kidney dysfunction even before the initiation of risk-adapted treatment with hydroxyurea or anagrelide and that the decision to prescribe any of these drugs is associated with an elevation in creatinine levels. Individuals with ET and concomitant HTN had higher creatinine concentrations before the commencement of treatment (*p* < 0.001) and experienced a significant rise in creatinine levels after cytoreduction with hydroxyurea or anagrelide was prescribed (0.91 mg/dL pre-treatment versus 0.96 mg/dL post-treatment, *p* < 0.001). ET patients receiving antihypertensive agents (OR = 2.20, 95% CI: 1.26–3.85, *p* = 0.006) and anagrelide (OR = 13.01, 95% CI: 6.27–27.01, *p* < 0.001), as well as harboring CALR mutations (OR = 1.95, 95% CI: 1.10–3.45, *p* = 0.02), were more likely to experience kidney dysfunction [13]. Interestingly, the post-mortem evaluation of the kidneys in 57 individuals with MPNs did not delineate proof of HTN or T2DM-related nephropathy in the subjects who exhibited diffuse glomerulosclerosis [37].

Furthermore, there seems to be a crosstalk between HTN and the serum concentrations of uric acid in patients living with MPNs. Lucijanic et al. analyzed 125 PMF and 48 SMF subjects in comparison with 30 healthy controls and reported that the PMF/SMF subjects who had high vs. low uric acid levels at diagnosis were more likely to suffer from HTN (PMF: ~71% vs. ~52%, *p* = 0.03; SMF: ~65% vs. ~30%, *p* = 0.01) [38].

Moreover, patients with MPNs might exhibit a peculiar diurnal BP rhythm pattern as opposed to that of the general population with HTN. For example, Akdi et al. revealed that PV patients (*n* = 50) were more likely to present as non-dippers (~64% vs. ~37%, *p* = 0.007) and to display a lower nocturnal fall in SBP, DBP and MBP (−6.91 ± 8.92 mmHg vs. −11.65 ± 7.68 mmHg, *p* = 0.005; −11.31 ± 12.18 mmHg vs. −16.28 ± 12.03 mmHg, *p* = 0.042; −9.31 ± 10.41 mmHg vs. −14.06 ± 9.41 mmHg, *p* = 0.018) vs. hypertensive non-PV counterparts. Despite the fact that the average 24 h and daytime SBP and DBP were similar between the two groups, the nighttime SBP and DBP were elevated in the PV subjects (125.32 ± 17.24 mmHg vs. 118.93 ± 12.20 mmHg, *p* = 0.034; 73.74 ± 12.16 mmHg vs. 69.49 ± 8.52 mmHg, *p* = 0.044, respectively). Irrespective of the presence of PV, the nocturnal falls in SBP (r = 0.305, *p* = 0.002 and r = 0.354, *p* < 0.001), DBP (r = 0.197, *p* = 0.048 and r = 0.236, *p* = 0.017) and MBP (r = 0.246, *p* = 0.013 and r = 0.293, *p* = 0.003) were positively associated with hemoglobin and hematocrit levels, respectively [39]. However, an age-matched and sex-matched Polish case-control study did not identify significant differences in terms of office or 24 h SBP, DBP, heart rate or use of antihypertensive agents in PV vs. healthy counterparts. However, the investigators detected signs of systolic and diastolic dysfunction of the heart by the assessment of several echocardiographic parameters, i.e., septal and lateral systolic velocities (8.7 ± 1.3 vs. 7.2 ± 2.4, *p* = 0.04 and 9.3 ± 1.2 vs. 7.7 ± 2.4, *p* = 0.04, respectively) and global longitudinal, circumferential and radial strains (−20.1 ± 4.3% vs. −18.1 ± 3.1%, *p* = 0.01; −19.7 ± 1.1% vs. −16.7 ± 2.7%, *p* = 0.001; 37.5 ± 8.7% vs. 29.6 ± 12.8%, *p* = 0.05, respectively) were elevated in the healthy controls, whereas the isovolumic relaxation time was increased in the patients with PV (110.9 ± 24.9 ms vs. 83.5 ± 12.9 ms, *p* = 0.0001). In addition, the authors noted that, in PV, hemoglobin levels were negatively associated with the global longitudinal and circumferential strains (β = −0.488, *p* = 0.0001; β = −0.537, *p* = 0.005), whereas the hematocrit value was positively linked only with the global longitudinal strain (β = 0.408, *p* = 0.001) and the red blood cell count was negatively correlated with the isovolumic relaxation time (β = −0.463, *p* = 0.05) [5].

Jóźwik-Plebanek et al. (2020) analyzed the BP variability, the activity in the adrenergic nervous system and the subclinical target organ damage in 20 individuals diagnosed with PV versus 20 age- and sex-matched controls. HTN affected 75% of the study group which displayed lower 24 h SBP and 24 h DBP (*p* = 0.003 and *p* = 0.01, respectively) versus comparators. All other office or ambulatory BP measurements were similar in PV and the controls. In terms of biochemical markers, PV was characterized by decreased metanephrine and aldosterone in the plasma (*p* < 0.001 and *p* = 0.008, respectively) and increased potassium levels (*p* < 0.001), as well as reduced free normetanephrine, metanephrine and norepinephrine in the urine (*p* = 0.03, *p* = 0.007 and *p* = 0.03, respectively). The number of erythrocytes (*p* < 0.001), leukocytes (*p* = 0.001), platelets (*p* < 0.001), as well as hemoglobin (*p* < 0.001) and hematocrit (*p* = 0.02) were elevated in PV, whereas the mean corpuscular volume, mean corpuscular hemoglobin concentration and mean corpuscular hemoglobin were lower (*p* < 0.001 for all). In terms of retinal or kidney physiology, as well as cardiac ultrasound parameters, subjects with PV only exhibited reduced capillary blood flow in the retina (*p* = 0.08) which was also inversely associated with hemoglobin levels and erythrocyte numbers in these individuals (r = −0.57; *p* = 0.001 and r = −0.40, *p* = 0.02, respectively). In addition, aldosterone concentrations negatively correlated with the aforementioned red cell parameters (r = −0.33, *p* = 0.04 for both). When the activity in the adrenergic system was evaluated, the researchers discovered that, although HR baroreflex control was not different in the PV and healthy counterparts, daytime, nighttime and 24 h ABPM, in addition to muscle adrenergic nerve activity (*p* = 0.007 for bursts/min and *p* = 0.04 for bursts/100 heartbeats) were reduced in PV [40]. Interestingly, the occurrence of HTN in PV might be attributed to the crosstalk between the cell-free hemoglobin and nitric oxide concentrations in the plasma rather than to high hematocrit values. Apart from alterations of the complete blood count, the PV subjects exhibited a higher blood viscosity (*p* < 0.01), SBP (*p* < 0.05), DBP (*p* < 0.05), MAP (*p* < 0.05), cell-free hemoglobin (*p* < 0.01) and nitrite/nitrate (*p* < 0.01) versus healthy comparators especially when PV and HTN co-occurred. The researchers delineated that cell-free hemoglobin values were positively associated with MAP (r = +0.49, *p* < 0.05), nitrite/nitrate (r = +0.46, *p* < 0.05), hematocrit (r = +0.47, *p* < 0.05) and blood viscosity (r = +0.39, *p* < 0.05). Isovolemic erythrocytapheresis decreased several of the biochemical parameters; however, it barely influenced the cell-free hemoglobin—hematocrit, cell-free hemoglobin—blood viscosity associations which continued to display a positive trend line and remained statistically significant (*p* < 0.05) [41].

The main findings of this subsection are summarized in Table 1.

### 3.2. Hypertension and Thrombosis in Myeloproliferative Neoplasms

MPN patients are prone to develop both thrombotic and hemorrhagic complications. Carobbio et al. analyzed the development of arterial and venous thrombosis in ET and discovered that the presence of HTN, T2DM or smoking (at least one CVRF) was a predictor of both major thrombotic events (HR = 1.56, 95% CI = 1.03–2.36, *p* = 0.038). In particular, HTN, T2DM or tobacco use were predictors of major arterial thrombotic events (HR = 1.91, 95% CI = 1.19–3.07, *p* = 0.007), namely acute myocardial infarction, ischemic stroke, cerebral transient ischemic attacks or peripheral arterial thrombosis, but not of the occurrence of major venous thrombosis (HR = 0.77, 95% CI = 0.33–1.83, *p* = 0.556), namely venous thromboembolism [42]. Likewise, Buxhofer-Ausch et al. sought to examine the impact of lifestyle factors on the development of thrombotic events in 105 ET and 62 early PMF subjects. HTN was similarly detected in both subgroups (50% vs. 44%, *p* = 0.48). HTN was a risk factor for the development of thrombosis (univariate model: HR = 3.43, range 1.12–10.52, *p* = 0.03; multivariate model: HR = 3.33, range 0.90–12.29, *p* = 0.07) and, in particular, arterial thrombosis (only in the univariate model: HR = 3.76, range 1.05–13.48, *p* = 0.04; multivariate model: HR = 2.79, range 7.06–11.02, *p* = 0.14) in ET. However, it did not have any impact on the occurrence of venous thrombosis (HR = 2.09, range 0.19–23.11, *p* = 0.55) in ET [43]. Similarly, Pósfai et al. investigated the role of CVRFs in the development of thrombosis in 101 ET patients hospitalized in a Hungarian hospital and found out that HTN (the most common comorbidity in the study group, 46.5%) was not linked to the occurrence of thrombosis in the logistic regression analysis. However, when the co-existence of two or more CVRFs, out of HTN, dyslipidemia, diabetes or smoking, was noted, it was linked to the development of thrombotic events (*p* = 0.02). Overall, the probability of thrombosis-free survival was lower in ET patients with ≥1 CVRF vs. those without CVRFs (*p* = 0.01) and in ET patients with one CVRF vs. ≥2 CVRFs (*p* = 0.002) [44]. Despite these findings, in a previous paper, Pósfai et al. identified HTN as a predisposing factor (*p* = 0.001) to the development of thrombotic complications in 128 ET female patients of whom ~55% (*n* = 70) had elevated BP values. Similarly, the presence of ≥2 CVRFs was linked with an elevated probability of suffering a thrombotic event in women diagnosed with ET (RR = 4.728, 95% CI 1.312–17.040, *p* = 0.01) [45]. Horvat et al. also demonstrated that HTN and the presence of ≥1 CVRF as risk factors for thrombotic events (OR = 2.8, 95% CI 1.6–5.0, *p* < 0.001; OR = 3.2, 95% CI 1.7–6.3, *p* = 0.001, respectively), especially arterial thrombosis (OR = 3.3, 95% CI 1.7–6.3, *p* < 0.001; OR = 5.7, 95% CI 2.3–13.9, *p* < 0.001, respectively) in a sample of 258 MPN patients. On the one hand, the subgroup analysis based on the MPN subtype revealed that, in PV (*n* = 70) and PMF (*n* = 54), the presence of ≥1 CVRF but not HTN alone predicted the development of arterial thrombotic complications (OR = 7.9, 95% CI 1.0–64.9, *p* = 0.049; OR = 12.2, 95% CI 0.7–225.3, *p* = 0.044, respectively). On the other hand, both HTN and the presence of ≥1 CVRF were risk factors not only for overall thrombosis (OR = 3.8, 95% CI 1.6–8.7, *p* = 0.003; OR = 5.1, 95% CI 1.8–14.1, *p* = 0.001, respectively), but also for arterial (OR = 2.8, 95% CI 1.2–6.5, *p* = 0.021; OR = 3.9, 95% CI 1.4–11.1, *p* = 0.009, respectively) and venous (OR = 30.3, 95% CI 1.7–532.4, *p* < 0.001; OR = 17.1, 95% CI 1.0–300.8, *p* = 0.005) thrombosis separately in ET (*n* = 134) [46]. In Lekovic et al.’s assessment of 244 ET individuals diagnosed with ET of which ~58% had HTN, the development of both arterial and global thrombosis was associated with the presence of HTN (*p* = 0.01 and *p* = 0.001, respectively), CVRFs in general (*p* = 0.01 and *p* = 0.002, respectively) and with the number of CVRFs (*p* < 0.001 and *p* < 0.001, respectively) [47]. The same author group later pointed out that, in a cohort of 244 ET subjects who were followed-up for 7 years, CVRFs (HTN, T2DM and dyslipidemia) and the combination of CVRFs and tobacco use were less common in the patients who were still alive at the time of the analysis (~62% versus ~78%, *p* = 0.05 and ~21% versus ~41%, *p* = 0.01, respectively). Thus, the presence of CVRFs (HR = 2.33) and CVRFs + tobacco use (HR = 2.08) was linked with a shorter overall survival in ET. Based on these findings, the investigators proposed a novel assessment tool for the prognosis of ET, namely the Cardio-IPSET prognostic model which takes into consideration the following factors: age, history of thrombotic events, leukocyte count and the presence of CVRFs (HTN, T2DM, dyslipidemia and smoking). Considering that ~75% of deaths in ET were attributed to cardiovascular causes, the Cardio-IPSET instrument might play a key role in the prognostication of this MPN in the near future [48]. Likewise, Schwarz et al. stressed that, in individuals diagnosed with MPNs treated with anagrelide, HTN emerged as a predictor of overall thrombosis (*p* = 0.003), major thrombosis (*p* = 0.022) and arterial thrombosis (*p* < 0.001); however, it did not predict the occurrence of microvascular events or venous thrombotic events based on the univariate analysis. Moreover, in the multivariate regression analysis, HTN was the best predictor of arterial thrombotic events (OR = 1.813, 95% CI 1.295–2.538, *p* = 0.001) [49].

Accurso et al. (2020) also investigated the occurrence of thrombosis in MPNs in relationship with the presence of CVRFs. HTN was the most common cardiovascular comorbidity in 403 MPN subjects (165 PV and 238 ET cases), affecting ~64% of the individuals from both subgroups. An elevated percentage of PV vs. ET cases (~39% vs. ~27%, *p* = 0.014) experienced thrombotic complications. The presence of CVRFs was also associated with decreased survival in both PV (*p* = 0.014) and ET (*p* = 0.036) [50]. Cucuianu et al. (2006) analyzed the impact of CVRFs in 37 MPN cases (29 PV, 8 ET) and revealed that ~31% of the patients had HTN and that, in PV, the association of HTN, platelet count > 600,000 platelets/mmc and hematocrit > 55% was linked with a higher incidence of thrombotic events (*p* = 0.02) [51]. Interestingly, Barbui et al. (2017) pointed out that, among CVRFs, HTN impacts the incidence of thrombosis in low-risk PV (*n* = 525). Thrombosis-free survival was higher in low-risk PV patients who did not suffer from HTN (IR = 0.85, 95% CI 0.57–1.25 vs. IR = 2.05, 95% CI 1.34–3.14, *p* = 0.025). As compared to ET (*n* = 891), HTN was more prevalent in PV (OR = 1.38, *p* = 0.022) and BP values positively correlated with hematocrit levels [52]. Furthermore, Benevolo et al. (2021) recently reinforced, based on a study sample of 861 individuals, that HTN (HR = 1.77, 95% CI 1.03–3.06, *p* = 0.04) and a previous history of thrombosis (HR = 2.10, 95% CI 1.21–3.60, *p* = 0.01) elevate the risk of thrombotic complications in PV [53]. In addition, in the post-hoc multivariate analysis of the Evaluation of Anagrelide Efficacy and Long-term Safety study, a long-term study with a prospective observational design which recruited 3649 individuals diagnosed with high-risk ET of whom 34% had elevated BP (the most common CVRF in ET), HTN emerged as a predictor of both major hemorrhages (HR = 1.33, 95% CI 1.04–1.69, *p* = 0.02) and thrombohemorrhagic complications (HR = 1.69, 95% CI 1.02–2.79, *p* = 0.04) [54]. Cerquozzi et al. (2017) analyzed 587 cases of PV of whom 42% had HTN and depicted that the rate of arterial and venous thrombotic complications was elevated in subjects with elevated BP (52% vs. 38%, *p* = 0.004 and 44% vs. 30%, *p* = 0.009, respectively). Individuals with PV had a lower thrombosis-free survival (HR = 1.7, 95% CI 1.1–2.6, *p* = 0.02) in the univariate but not in the multivariate analysis [55]. Similarly, patients with any CVRF (HTN, T2DM, hypercholesterolemia, use of cigarettes) were at an elevated risk for thrombosis (OR = 14.9, 95% CI 2.5–87, *p* = 0.003) and had a lower thrombosis-free survival (~83% vs. 97%, *p* = 0.02) in a Spanish cohort of PMF/SMF subjects (*n* = 155) [56]. A Brazilian study conducted on 46 ET patients also confirmed the presence of an association between the aforementioned CVRFs and thrombosis (*p* = 0.01), namely arterial (*p* = 0.03) and not venous (*p* > 0.05) thrombotic complications [57].

Shih et al. analyzed the occurrence of thrombosis in 89 women diagnosed with ET and with/without clonal/polyclonal X-chromosome inactivation patterns. Thrombosis but not hemorrhage was more common in ET subjects with vs. without HTN (*p* = 0.002 and *p* = 0.287, respectively). After the adjustment for the presence of HTN and for age, the researchers concluded that the risk of thrombotic events was seven times more elevated in ET individuals with clonal X-chromosome inactivation patterns vs. those without [58].

However, not all studies reported an impact of HTN on the development of thrombosis in MPNs. Bucalossi et al. (1996) depicted a similar prevalence of HTN in 81 subjects with PV and ET with/without thrombotic complications [59]. In addition, in Landolfi et al.’s assessment of 1638 subjects from the European Collaboration on Low-Dose Aspirin in Polycythemia Vera (ECLAP), HTN did not emerge as a predictor for major/arterial/venous thrombosis, AMI, TIA, stroke or peripheral arterial thrombosis, although previously published analysis of the ECLAP cohort dating back to 2004 had reported the contrary [60,61]. Similarly, HTN did not impact prognosis, namely thrombosis-free survival and life expectancy in two Italian studies which recruited 187 and 100 patients diagnosed with ET, respectively [62,63]. In addition, Jantunen et al. evaluated 132 individuals diagnosed with ET and highlighted that cigarette use was a more common risk factor for thrombosis versus HTN (24.3% versus 20.5%). Moreover, male gender (*p* < 0.001) and tobacco consumption (*p* = 0.01) emerged as risk factors for thrombotic complications, whereas HTN did not (*p* = 0.34) [64].

Similarly, HTN was a more common occurrence in MPN individuals who developed an ischemic stroke versus those with transient ischemic attacks and it also emerged as a factor of prognosis in the recurrence of this thrombotic complication (HR = 4.24) [65]. Košťál et al. investigated the development of stroke and TIA based on data from a registry of 1445 MPN patients. The aforementioned complications were documented in 249 cases. HTN was a more common finding in the subgroup of individuals who experienced a stroke or a TIA (~53% vs. ~41%) and emerged as a risk factor for such complications based in the univariate analysis model (OR = 1.604, 95% CI = 1.219–2.111, *p* = 0.001) but not in the multivariate logistic models on data with imputed missing values (OR = 1.170, 95% CI 0.845–1.619, *p* = 0.344 for treated and untreated subjects; OR = 0.918, 95% CI = 0.55–1.534, *p* = 0.745 for subjects not receiving cytoreductive agents) [66]. De Stefano et al. (2018) analyzed 597 MPN cases with a history of stroke or TIA and discovered that the frequency of HTN was similar between the two patient subgroups (stroke vs. TIA = 57% vs. 52%, *p* > 0.05). However, HTN was discovered to act as an independent risk factor for the recurrence of ischemic stroke in MPNs (HR = 4.24, 95% CI 1.23–14.7). Cytoreduction was able to decrease the risk of stroke re-occurrence by 76% [67].

However, HTN is less frequent in MPN patients who experience CVST. Jiao et al. (2021) evaluated 91 individuals diagnosed with ET and discovered that HTN was more prevalent (~32% vs. ~4%, *p* = 0.003) in ET cases who did not develop CVST [68].

Interestingly, when compared to subjects with HTN, individuals diagnosed with MPNs display elevated concentrations of soluble *p*-selectin (*p* < 0.001), particularly if they harbor the *JAK2V617F* mutation (*p* = 0.006 between *JAK2V617F*-positive and *JAK2V617F*-negative cases), and D-dimers (*p* = 0.03), but similar soluble E-selectin, thrombin–antithrombin complexes, prothrombin fragments or antiphospholipid antibodies. However, soluble *p*-selectin levels were similar in MPN patients who experienced thrombotic events versus those who did not [69].

The main findings of this subsection are summarized in Table 2.

### 3.3. Hypertension Treatment Strategies in Myeloproliferative Neoplasms

Angiotensin-converting enzyme inhibitors (ACEIs) have been suggested as the first choice in the management of HTN in PV. Based on a secondary analysis of the European Collaboration on Low-Dose Aspirin in Polycythemia Vera (ECLAP) trial, Barbui et al. reported that the use of ACEIs in PV subjects with HTN can reduce the need for chemotherapy agents, as well as have an impact on clonal erythropoiesis. In the ECLAP trial, antihypertensive agents were prescribed alone or in combination in 647 of the 1638 PV cases enrolled. ACEIs were the most frequently prescribed HTN medication (*n* = 285), followed by calcium-channel blockers (*n* = 177), beta-blockers (*n* = 103) and/or diuretics (*n* = 96). Interestingly, individuals with PV who were prescribed ACEIs were less likely to require the use of chemotherapy agents. This finding was detected in the univariate model (OR = 0.78, *p* = 006) and later confirmed in the multivariable analysis (OR = 0.62; 95% CI, 0.44–0.88; *p* = 0.008). Moreover, median hematocrit values were similar in the ACEI-treated subgroup versus PV patients with HTN who received other medications in spite of the latter being administered more chemotherapy agents to control clonal erythrocytosis. However, thrombosis-free survival was similar between PV patients with/without ACEI treatment [70]. In addition, the use of ACEIs in PV seems to positively impact kidney function. In a retrospective assessment of 98 PV cases, patients who received ACEI treatment had a higher frequency of CVRFs (*p* < 0.001), HTN (*p* = 0.012) or diabetes (*p* = 0.035), as well as lower eGFR values (*p* = 0.046). However, treatment with ACEIs was linked to an improvement in kidney function, i.e., an increase in eGFR by at least 10% (OR = 6.97, 95% CI: 3.83–8.89, *p* = 0.008). Moreover, the use of ACEIs was also linked with an elevation in eGFR in high-risk PV (OR = 5.79, *p* = 0.016), as well as in PV subjects with CVRFs (OR = 3.88, *p* = 0.048) [71].

Gorokhovskaya et al. investigated the benefits of 10–40 mg lisinopril monotherapy administered once daily in the morning in subjects with 2nd or 3rd grade HTN and PV (*n* = 20) for 4 weeks. Following lisinopril prescription, there was a decrease in mean 24 h values for SBP (~↓12%; decrease encountered in 10% of the study group), DBP (~↓10%; decrease encountered in 20% of the study group), PP (↓25%) and of the double product (~↓15%), i.e., SBP multiplied by PP. BP variability and its 24 h profile was alleviated in 85% of the recruited individuals, whereas 65% of the study group achieved the target mean 24 h BP of <135/85 mmHg. Lisinopril treatment did not result in side effects, e.g., dry cough [72]. Rao et al. managed to reduce BP values in a successful fashion, from ~210/130 mmHg to ~130/70 mmHg, with the administration of a calcium-channel blocker, namely long-acting nifedipine [73].

The burden of HTN can also lead to changes in the administration of chemotherapy regimens to blood cancer patients. In an analysis of 330 subjects diagnosed with hematological malignancies (*n* = 43, ~13% had MPNs), HTN was the most common comorbidity encountered (*n* = 96, ~35%) and was a factor associated with changes in treatment options (*n* = 62, ~19%) in both the univariate (*p* = 0.01) and the multivariate (*p* = 0.003) analysis. Moreover, its presence was a predictor of treatment modifications (OR = 2.743, 95% CI 1.397–5.386, *p* = 0.003) [74].

### 3.4. Drug-Induced Hypertension and the Effect of Hematological Treatment on Hypertension as a Cardiovascular Risk Factor

Although ruxolitinib leads to an elevation in the BMI via the inhibition of JAK2/STAT3 (which mediates leptin receptor signaling in the hypothalamus), it has no effect on SBP nor DBP [75]. Sapre et al. (2019) also evaluated the actions of RUX administration on cardiometabolic health. HTN was detected in ~64% of MPN patients at baseline and in ~69% of the study group after 72 weeks of RUX prescription, but the difference was not statistically significant (*p* > 0.05). RUX did not alter DBP (72 mmHg vs. 70 mmHg, *p* > 0.05) but it increased SBP (124 mmHg vs. 129 mmHg, *p* = 0.031) [76].

Drug-induced HTN is a frequently underdiagnosed entity in medical practice. Secondary HTN can often develop as an adverse effect of pharmacological treatment. HTN was reported as a non-hematological AE in 9% (*n* = 7) of the 74 PV subjects who were prescribed RUX in the RESPONSE-2 trial. HTN was more commonly noted as a grade 3–4 (grade 3: 5%, *n* = 4; grade 4: 1%, *n* = 1) versus grade 1–2 (3%, *n* = 2) AE in the RUX group, whereas PV subjects receiving the best available treatment experienced HTN only as a grade 3 AE (4%, *n* = 3) [77]. For example, the use of ruxolitinib –decitabine combination treatment in accelerated or blast phase MPNs in 25 subjects resulted in several hematological and non-hematological adverse effects, including HTN (*n* = 3; 12% of the study group). HTN was more commonly a grade 1/2 (*n* = 2; 8% of the study group) rather than a grade 3+ adverse event (*n* = 1; 4% of the study group) [78]. Previously, Rampal et al. (2018) had reported that the RUX–decitabine combination in 21 MPNs in the accelerated or blast phase caused HTN as a grade 3 AE to only 2 patients (9.5%): 1 receiving 25 mg of RUX twice daily and 1 receiving 50 mg of RUX twice daily both in combination with decitabine (20 mg/m^2^ daily for 5 days) [79]. Similarly, RUX–danazol administration in 14 subjects with PMF/SMF caused HTN as a non-hematological toxicity (*n* = 1, 7.1%) in a multicentric phase 2 study [80]. Griesshammer et al. (2018) recorded HTN as the most common grade 3/4 adverse event in patients exposed to RUX (all grades: 8.3/100 patient-years of exposure; grade 3/4: 6.8/100 patient-years of exposure), in the RUX cross-over arm (all grades: 9.0/100 patient-years of exposure; grade 3/4: 6.0/100 patient-years of exposure) or best available treatment arm (all grades: 5.6/100 patient-years of exposure; grade 3/4: 5.6/100 patient-years of exposure) in the RESPONSE-2 trial [81]. In addition, Kiladjian (2018) reported that, after cross-over from IFN to RUX, no patient out of 13 recruited was reported to have HTN in the RESPONSE trial, whereas only 1 out of 8 patients recruited developed HTN as a grade 1/2 adverse event in the RESPONSE-2 trial (13.5/100 patient-years of exposure) [83]. In the 5-year follow-up of the PV patients enrolled in RESPONSE, Kiladjian (2020) later assessed the presence of HTN as an adverse event in the best available treatment (*n* = 111), cross-over (*n* = 98) and RUX (*n* = 110) arms. HTN was registered as an adverse event (all grades, grade 1/2 and grade 3/4) primarily in the best available treatment (5.4, 4.0 and 1.4 per 100 patient-years of exposure, respectively), followed by the cross-over (4.5, 3.6 and 0.9 per 100 patient-years of exposure, respectively) and the RUX (4.0, 3.5 and 0.5 per 100 patient-years of exposure) arm, respectively [84]. AT9283, a small molecule inhibitor of Aurora kinases A and B, c-ABL, *JAK2* and other kinases caused HTN as a dose-limiting toxicity in an open-label, dose-escalation study investigating this agent in the treatment of refractory AML and advanced PMF/SMF [85].

Rambaldi et al. (2021) investigated the use of givinostat in PV based on data from a compassionate use program and phase 1/2 trials (*n* = 50) and discovered that only one patient developed HTN as a grade 3 adverse event following the administration of this drug. At the baseline, 58% of the study group (*n* = 29) suffered from controlled HTN [86].

Gotic et al. (2020) evaluated the safety of anagrelide vs. hydroxyurea administration in 146 ET subjects enrolled in a phase 3b multicentric randomized, open-label, non-inferiority study. HTN was the most frequent vascular disorder registered as treatment-emergent adverse events (anagrelide: *n* = 9, 11.8% vs. hydroxyurea: *n* = 1, 1.4%) [87]. In a retrospective evaluation of the Italian ET registry (Registro Italiano Trombocitemia, RIT), HTN was also reported as an AE to anagrelide administration in 3.5% (*n* = 8) of 232 subjects treated with a platelet-lowering agent [88]. However, Tortorella et al. (2015) reported that an in-depth cardiovascular assessment prior to anagrelide treatment initiation does not predict the development of cardiovascular adverse events in ET. In their prospective study, they evaluated 55 ET patients of whom 25.5% (*n* = 14) also suffered from HTN. After cardiovascular screening was performed, 54 subjects were deemed eligible to receive anagrelide treatment and treatment was finally started in 38 ET cases of whom only 5.3% (*n* = 2) of patients registered HTN as a cardiovascular adverse event. However, anagrelide-induced HTN did not require drug discontinuation, nor referral to a cardiologist, and was successfully managed by the attending hematology physician [89].

## 4. Discussion

### 4.1. Hypertension Is the Most Common Comorbidity in Myeloproliferative Neoplasms

Hypertension (HTN) is the most common comorbidity in patients diagnosed with myeloproliferative neoplasms (MPNs). The high prevalence of HTN is seen across the spectrum of various MPNs (ET, PV, PMF, SMF.). The *JAK2V617F* mutation is present in a vast majority of MPNs and may be implicated in the pathogenesis of both systemic and pulmonary arterial hypertension (PAH). The *JAK2V617F* mutation leads to the upregulation and accumulation of neutrophils within the pulmonary arterial vasculature causing PAH [90]. In addition, the *JAK2V617F* mutation can cause arterial elastic laminal degradation and may play a role in the pathogenesis of systemic HTN [4].

HTN is also the most prevalent comorbidity in patients with COVID-19 with MPNs; however, it is not associated with a significant increase in morbidity and mortality. HTN is more commonly seen in MPN patients prescribed lipid-lowering drugs, e.g., statins. This represents a correlation of HTN along with other CVRFs, e.g., dyslipidemia, amongst others in patients with MPNs [32]. Likewise, the risk of 5-year vascular disease in patients with MPNs is significantly elevated in comparison to the general population [90]. The presence of HTN in MPNs is associated with an altered neurohormonal axis in comparison to individuals with HTN without MPNs. Specifically, the levels of aldosterone, serum-free epinephrine and muscle sympathetic nervous activity (measured via microneurography) are significantly lower. The decreased sympathetic nervous activity and lower levels of aldosterone and free epinephrine likely represent a reflexive response to decreasing BP caused by MPNs. Indeed, the dysfunctional proliferation of myeloid cells is most likely contributing to arteriolar dysfunction; however, further studies are needed on the pathophysiology [10,11].

The presence of HTN in MPNs was also found to be associated with an increased risk for organ dysfunction. Adverse outcomes include glomerulopathy, carotid artery injury, central retinal artery occlusion, renal failure and systemic thrombosis. ET carries the most significant risk for thrombotic events and kidney dysfunction in comparison to other MPNs [12,13]. MPNs are also associated with altered cardiac physiology and neural control of BP. Patients with PV exhibited increased isovolumic relaxation time assessed via echocardiography. Systolic velocities were also significantly lower in PV [5]. Increased hematocrit in PV causes a subsequent increase in blood viscosity as described by Poiseuille’s equation. The effects on the underlying hemodynamics include increased isovolumic relaxation to allow for adequate cardiac chamber filling (preload) and reduced systolic velocities. HTN in MPNs, e.g., PV, is associated not only with increased SBP, DBP and MAP, but also cell-free hemoglobin and nitrate levels. Increased cell-free hemoglobin has been implicated in increased systemic vascular resistance via the scavenging of nitric oxide; however, further studies are still needed to validate the exact mechanism of action [91].

### 4.2. Relationship of Hypertension and Other Cardiovascular Risk Factors on Thrombotic Events in Myeloproliferative Neoplasms

The presence of CVRFs (T2DM, HTN, smoking, etc.) is readily correlated with the development of systemic thrombotic phenomenon, including myocardial infarction, peripheral arterial thrombosis and transient ischemic attacks. The presence of HTN is more closely related to thrombotic phenomenon within the arterial rather than venous vasculature [42,43,49,56,57]. The presence of > 1 CVRF in addition to HTN has shown a greater risk of arterial thrombosis [46,47]. HTN is also correlated with hemorrhagic and thrombohemorrhagic adverse events in patients with ET [54]. The presence of the clonal X-chromosome inactivation pattern was closely correlated with an increased risk of thrombosis than those without while controlling for factors such as HTN [58]. Male gender and smoking rather than HTN was a more common risk factor for thrombotic events in some investigations [62,64].

Although many studies have shown a correlation between HTN and thrombotic events, several studies have shown that HTN is not a predictor for arterial/venous thrombotic events [59,60,61,62,63]. The presence of HTN in MPNs is also closely associated with CNS thrombotic phenomenon, including both stroke and TIA [65,66,67]. Cytoreduction decreases the risk of stroke recurrence by 76% [67]. Interestingly, HTN presence is negatively correlated with CVST occurrence [68]. This finding coincides with the aforementioned lack of correlation of HTN with venous thrombotic events.

### 4.3. Hypertension Treatment Strategies in Myeloproliferative Neoplasms

Angiotensin-converting enzyme inhibitors (ACEIs) are regarded as the first choice in the management of HTN in PV [70]. Angiotensin II helps in the erythropoietin-mediated proliferation of red cell precursors which in turn raises the hematocrit and causes hypertension [92]. ACEIs are used in randomized control trials to reduce hematocrit in post-renal-transplant patients [93,94]. Similar to this hypothesis, some evidence indicated that ACEIs may have a role in the management of PV [95], leading to the normalization of the blood cell count and partial remission of heavy proteinuria [96]. Barbui et al., in the ECLAP trial, have reported that the use of ACEIs in PV patients not only reduced the use of chemotherapy agents (compared to other antihypertensive agents used in this trial, OR = 0.62; 95% CI, 0.44–0.88; *p* = 0.008) but also had an impact on the clonal erythropoiesis [70]. However, the use of ACEIs did not alter thrombosis-free survival. In another retrospective study by Krecak et al., treatment with ACEIs in PV patients was associated with an improvement in kidney function, evidenced by an increase in eGFR by at least 10% (OR = 6.97, 95% CI: 3.83–8.89, *p* = 0.008). Moreover, the use of ACEIs was also linked with an elevation in eGFR in high-risk PV (OR = 5.79, *p* = 0.016), as well as in PV subjects with CVRFs (OR = 3.88, *p* = 0.048) [71]. These inferences can be of potential interest, as low eGFR in all the three classical MPNs are associated with an increased risk of thrombotic events [97,98]. Rao et al. had shown the successful treatment of severe hypertension in MPNs with *JAK2* mutation using a long-acting calcium-channel blocker, i.e., nifedipine [73]. HTN, being the most common comorbid condition in hematological malignancies, often changes the treatment options. In a retrospective chart review by Najman and colleagues, HTN was not only an associated risk factor of hematological malignancies but also an independent predictor of treatment modification in these cases [74].

### 4.4. Drug-Induced Hypertension and the Effect of Hematological Treatment on Hypertension as a Cardiovascular Risk Factor

Certain drugs used in the treatment of MPNs are associated with HTN and other CVRFs. A widely used drug, ruxolitinib, is a JAK2/STAT3 inhibitor that causes hyperlipidemia and increases the body mass index. Sapre et al. had shown that there is no effect of ruxolitinib on blood pressure [76]. Contrary to this, in the RESPONSE-2 trial, 9% of PV patients who were receiving ruxolitinib had HTN and it was reported as a non-hematological adverse effect [77]. A similar result was shown by Mascarenhas and colleagues where 12% of the study participants treated with a combination of ruxolitinib and decitabine had HTN [78]. Other studies also supported the occurrence of drug-induced HTN at various degrees of severity [80,81,82,83]. In an insightful study by Molle et al., it was inferred that subjects who gained > 2% pre-treatment body weight within 60 days of the initiation of treatment were more likely to continue gaining weight and become overweight [75]. In the same study, it was identified that ruxolitinib interferes with post-prandial leptin signaling and thereby causes hyperphagia [75]. The *JAK2* gene is associated with smooth muscle, cardiac function and endothelial functions. Pharmacological inhibition of this gene has been reported to alter the expression of endothelial NOS [8,9], which in turn can potentiate the rise in BP. Hence, dietary counseling and lifestyle modifications should be advised in all patients who are started on this drug.

Another drug, AT9283, a small molecule inhibitor of Aurora kinases A and B, c-ABL, *JAK2* and other kinases, caused HTN as dose-limiting toxicity when used in the treatment of refractory AML and advanced PMF/SMF [84]. In another study, Gotic et al. evaluated the safety of anagrelide vs. hydroxyurea administration in 146 ET subjects and discovered that HTN was the most frequent vascular disorder (anagrelide: *n* = 9, 11.8% vs. hydroxyurea: *n* = 1, 1.4%) [86]. Another Italian study reported that 3.5% of patients on anagrelide developed HTN [87]. However, Tortorella et al. reported that in-depth cardiovascular evaluation before starting anagrelide treatment does not predict the development of cardiovascular adverse events in ET patients [88]. Moreover, anagrelide-induced HTN does not warrant discontinuation of the drug and can be managed successfully by hematologists without a further referral [88].

### 4.5. Caveats and Future Research Directions in Unravelling the Interplay between Hypertension and Myeloproliferative Neoplasms

To our knowledge, the present manuscript is the first systematic review to examine the interplay between primary arterial HTN and drug-induced HTN in MPNs. Based on our findings, we can conclude that HTN is the most common comorbidity in the aforementioned blood cancers, influencing not only the risk of thrombosis and the management strategies, but also the survival of patients who are diagnosed with both these disorders.

However, future research should also focus not only on understanding the clinical relevance of the association between MPNs and HTN, but also on molecular assessments exploring their interplay. For example, genomic-directed stratification, preferably in the setting of randomized controlled trial design and enrolment, but also of cohort investigations, might aid us in deciphering the complex crosstalk between CVRFs, cardiovascular disease and MPNs. In these hematological malignancies, cardiovascular disease has mainly been linked to the presence of the *JAK2V617F* mutation which is the most frequently encountered genetic alteration in MPNs. Thus, a potential objective of upcoming investigations should be based on genomic-driven stratifications in order to depict the relationship of other genetic changes in MPNs with HTN. For example, little is known today regarding the contribution to cardiovascular disease onset of other somatic mutations recognized in MPN subjects, i.e., mutations in the calreticulin (*CALR*) or myeloproliferative leukemia protein (*MPL*) driver genes, or about the interplay between HTN and genetic alterations in epigenetic regulators, e.g., DNA methyltransferase 3 alpha (*DNMT3A*), tet methylcytosine dioxygenase 2 (*TET2*), additional sex comb-like transcriptional regulator 1 (*ASXL1*), isocitrate dehydrogenase 1 or 2 (*IDH1/2*) genes [14,15,99,100]. The use of genomic stratification is of paramount importance in hematology and oncology, where the application of precision medicine and individualized approaches can aid better prognostication and treatment selection, including targeted treatments and candidate selection for bone marrow transplantation. For example, patients with MPNs can benefit from the use of next-generation deoxyribonucleic acid (DNA) sequencing-based gene panels or liquid biopsy-based tools that are able not only to stratify patients based on genetic alterations, but also predict the risk of thrombosis, progression to secondary myelofibrosis (PV-MF or ET-MF), transformation into acute myeloid leukemia (AML) and death [99,101,102,103]. However, a main limitation of MPN management worldwide is that genomics, proteomics and other sophisticated laboratory studies are not available everywhere in the world, largely due to the elevated costs and requirement for trained personnel to implement and handle such experiments. Therefore, clinical judgment combined with morphology (cytology and, most importantly, histopathology) studies and genetic testing for the *JAK2*, *CALR* and *MPL* mutations remain mandatory in the accurate diagnosis of MPNs globally [104]. Hence, the assessment and correct targeting of CVRFs, and in particular of HTN, are a must in the management of MPNs, and we believe that our systematic review has provided insightful information regarding the impact of this comorbidity in the aforementioned myeloid malignancies.

The pathogenesis of MPNs is complex and their onset requires the contribution of an intricate network of molecular mechanisms, as highlighted in Figure 2. Interestingly, Solimando et al. have hypothesized that clonal hematopoiesis of indeterminate potential (CHIP) can contribute to the development of chronic inflammation that hijacks the bone marrow microenvironment ecosystem and can eventually elevate the risk of MPN development [105]. The interplay between CHIP and inflammation has been demonstrated to lead not only to the development of hematological malignancies, e.g., MPNs, myelodysplastic syndromes, de novo or secondary AML, but also to systemic effects and the onset of cardiometabolic disorders, e.g., HTN, T2DM, cardiovascular disease etc. [15,100,105]. Thromboinflammation and its crosstalk with somatic mutations (*JAK2V617F* in particular), epigenetic regulators, splice factor modifiers and oxidative stress are well-established molecular events that drive the occurrence of thrombotic events, as well as phenotypic transformation to PV-MF or ET-MF and eventually accelerated/blastic phase MPNs. Consequently, future approaches to treatment targeting in MPNs and acute leukemia should also concentrate on and exploit the antioxidant and anti-inflammatory properties of current MPN medication, e.g., ropeginterferon alpha-2b or *JAK1/2* inhibition, as well as potentially synthesize novel compounds from natural products that exhibit these properties when available [106,107,108,109,110,111]. Moreover, upcoming investigations should also aim to dissect the relationship between CHIP and MPNs in order to identify CHIP individuals at risk of MPN development, and in particular those at higher risk to eventually be diagnosed with secondary AML [112,113].

Our paper has several strengths and limitations. Firstly, to our knowledge, it is the first systematic review to examine the relationship between MPNs and primary arterial HTN, as well as drug-induced HTN. As the assessment was based on data extracted from publications available in PubMed/MEDLINE, Web of Science and SCOPUS, we believe the presented data are robust. The main limitations of our research are the impossibility to compute a meta-analysis due to the heterogeneity of the data, as most of the studies were observational in design, and the fact that HTN was not necessarily a primary outcome of the investigated manuscripts. However, we are confident that this systematic review sheds more light on the association of HTN and MPNs, arguing for a better CVRF assessment and management in subjects suffering from MPNs.

## 5. Conclusions

Our systematic review highlights that HTN is the most common comorbidity encountered in MPNs, with an impact on both the occurrence of thrombosis and survival. Moreover, drug-induced HTN remains a challenge in the management of MPNs. Further research should investigate the characteristics of patients with MPNs and HTN, as well as to clarify the contribution of HTN to the development of thrombotic complications, survival and management.

## Figures and Tables

**Figure 1 biomedicines-11-00388-f001:**
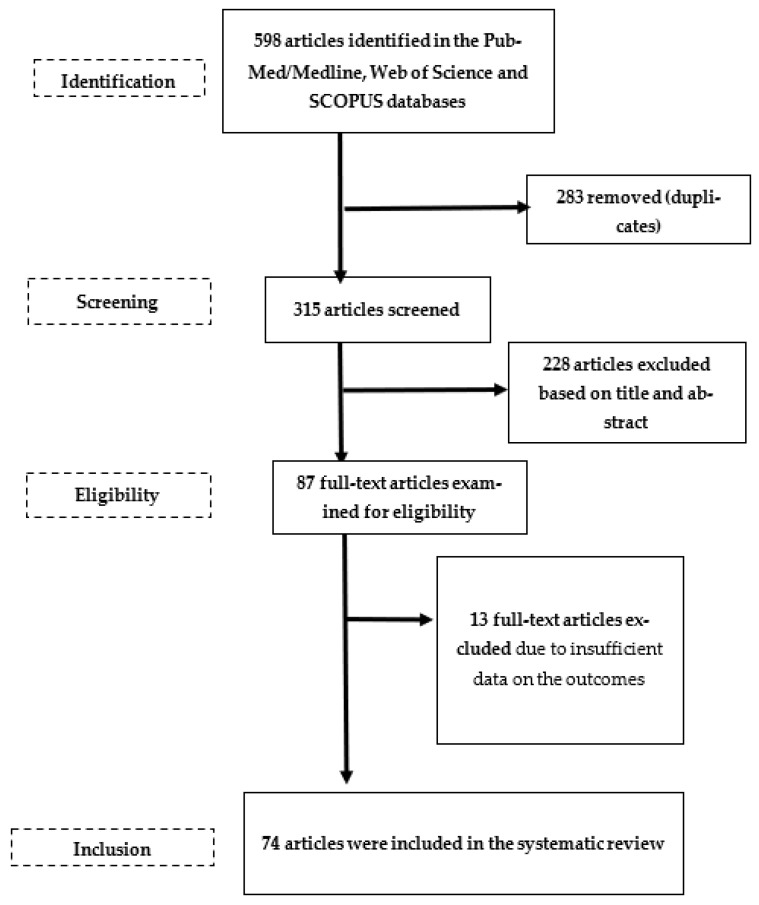
Flowchart depicting the study selection and inclusion processes for the current systematic review.

**Figure 2 biomedicines-11-00388-f002:**
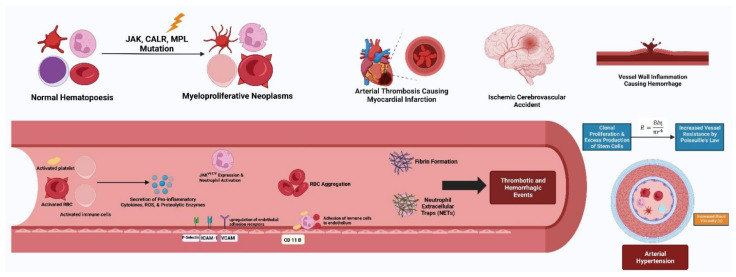
The pathogenesis of myeloproliferative neoplasms is multifaceted and often includes genetic mutations involving the *JAK2*, *CALR* and *MPL* genes among others. Abnormal clone-derived blood cells (i.e., erythrocytes, platelets and leukocytes) drive prothrombotic features. The secretion of pro-inflammatory cytokines and reactive oxygen species disrupts normal endothelial function and leads to increased expression of endothelial adhesion receptors. The release of tissue factor from activated platelets aids in the formation of fibrin clots and DNA released from activated neutrophils forms neutrophil-extracellular traps (NETs). Additionally, red blood cell (RBC) aggregation occurs through biochemical changes in the RBC membrane. The culmination of these maladaptive events causes increased blood viscosity, vascular resistance and arterial hypertension through Poiseuille’s law.

**Table 1 biomedicines-11-00388-t001:** Main results of the studies regarding the epidemiology of HTN in MPNs.

Author and Year	Study Location	MPN Subtype	Number of Patients	Main Results
Mitra et al. (2013) [20]	USA	PMF, SMF	180	↑ BP in 52% of PMF/SMF patients with splenomegaly, 50% in those without HTN = most common comorbidity in PMF/SMF
Parasuraman et al. (2018) [21]	USA	PV	7718	HTN = most prevalent comorbidity (~72%) (*n* = 5531) of PV patients enrolled in the United States Veterans Health Administration database
Grunwald et al. (2018) [22]	USA	PV	2510	HTN = most prevalent comorbidity (~71%) (*n* = 1772) of PV subjects enrolled in The Prospective Observational Study of Patients With Polycythemia Vera in US Clinical Practices (REVEAL)
Paranagama et al. (2018) [23]	USA	PV	2856	HTN in ~57 (*n* = 1630) of PV cases ↑ BP more prevalent in high-risk vs. low-risk PV (~65% vs. ~43%, *p* < 0.001)
Accurso et al. (2020) [24]	Italy	PV, ET, PMF, SMF	603	↑BP = #1 CVRF in MPNs ↑ BP in ~63% of PV, ~64% of ET, ~15% of pre-fibrotic PMF, ~63% of PMF/SMF
Yacoub et al. (2021) [25]	USA	ET	1207	↑ BP in 56% of the 1207 ET Myelofibrosis and Essential Thrombocythemia Observational STudy (MOST), HTN #1 comorbidity in ET
Jain et al. (2021) [26]	India	PV	52	HTN #1 comorbidity in PV (*n* = 24/52 patients, 46%)
Yap et al. (2018) [27]	Malaysia	PV, ET, PMF	1010	association of HTN with *JAK2V617F* (OR = 1.688; 95% CI = 1.234–2.310; *p* = 0.001) and with the diagnosis of PV and ET HTN #1 comorbidity in MPN subjects, i.e., PV > ET > PMF > MPN unclassifiable > hypereosinophilic syndrome (~58% vs. ~42% vs. ~41% vs. ~36% vs. ~33%, *p* < 0.001)
Shah et al. (2021) [28]	Pakistan	PV	51	~90% of PV cases display HTN HTN associated with patients’ age (*p* = 0.033)
Mancuso et al. (2020) [29]	Italy	ET	233	HTN #1 CVRF in ET ~40% of ET cases display 1 CVRF, whereas ~35% have multiple CVRFs IPSET-thrombosis score calculation reclassifies several ET cases from low-risk of thrombotic events to intermediate or high-risk thrombotic complications occur mainly in ET with associated CVRFs
Fattizzo et al. (2020) [30]	Italy	NS	16	HTN #1 comorbidity (56%) in myeloid cancers (50% of which were MPNs or MPN/MDS) who developed COVID-19 HTN had no impact on survival in this setting
Morrissey et al. (2020) [31]	UK	NS	NS	HTN #1 comorbidity (41%) in COVID-19 patients with malignant (including MPNs) and benign blood disorders of whom 55% died HTN not linked with adverse outcomes
Delluc et al. (2018) [32]	France	PV, ET	192	HTN #1 comorbidity in MPNs (57.3%) HTN more likely in PV and ET cases on statin treatment (~85% vs. ~51%, *p* < 0.001)
Palandri et al. (2009) [33]	Italy	ET	386	HTN occurred in 48% of ET cases HTN independently predicted poor survival in ET
Wojcicki et al. (2016) [10] Lewandowski et al. (2017) [11]	Poland	PV	20 12	↑ BP in ¾ of PV subjects when examined in the investigator’s offices or on ambulatory BP measurements less notable ↓ in BP during the night ↓ heart rate values per 24 h (heart rate during the day was particularly ↓ in PV) ↓ microneurography-evaluated muscle sympathetic nervous activity (muscle activity per minute particularly ↓ in PV and positively associated with the heart rate during the night, r = 0.617, *p* < 0.05) ↓ serum free epinephrine, ↓ aldosterone concentrations (*p* = 0.048) in HTN + PV versus HTN non-PV patients no differences regarding the morphology of the retina between the two groups ↓ decreased retinal microperfusion (*p* = 0.032) in PV negatively associated with number of erythrocytes in PV (r = −0.44, *p* = 0.012) ↑ resistive index of the kidney arteries in PV versus non-PV HTN (*p* = 0.033)
Maruyama et al. (2019) [34]	Japan	PMF	1	HTN may aggravate MPN-related glomerulopathy via nephrosclerosis with development of polar vasculosis and exudative lesions
Patino-Alonso et al. (2021) [35]	Spain	PV, ET, PMF	57	similar values for SBP and DBP between MPNs (*n* = 57) and controls ↑ risk of carotid artery injury in MPNs (OR = 2.382, 1.066–5.323, *p* = 0.034) ↑ albumin-to-creatinine ratio in MPNs
Rana et al. (2021) [36]	Australia	NS	1	MPN and HTN association can contribute to the occlusion of the central artery of the retina and present as sudden loss of vision
Gecht et al. (2021) [37]	Germany	PV, ET, PMF, SMF	1420	49%o of 1420 MPN cases had HTN ↑ BP more common in PV vs. ET or vs. PMF/SMF (~63% vs. ~46% and ~63% vs. ~39%, respectively, *p* < 0.0001) MPN subjects have eGFR = 60–89 mL/min/1.73 m^2^ (~57%), followed by eGFR < 60 mL/min/1.73 m^2^ (~22%) and eGFR ≥ 90 mL/min/1.73 m^2^ (~21%) more PMF/SMF subjects had eGFR < 60 mL/min/1.73 m^2^ vs. PV or vs. ET (~28% vs. ~20%, *p* = 0.005 for both) HTN = risk factor (OR = 2.419, 95% CI 1.879–3.114, *p* < 0.0001) for kidney dysfunction, particularly in subjects with eGFR < 60 mL/min/1.73 m^2^ vs. eGFR ≥ 90 mL/min/1.73 m^2^ (OR = 6.220, 95% CI 1.501–25.779, *p* = 0.01) and vs. eGFR = 60–89 mL/min/1.73 m^2^ (OR = 3.429, 95% CI 1.207–9.739, *p* = 0.02) in the univariate regression analysis HTN = risk factor for kidney dysfunction (OR = 2.004, 95% CI 1.440–2.789, *p* < 0.0001) following multiple regression analysis HTN = risk factor for thrombotic (OR = 1.838, 95% CI 1.398–2.418, *p* < 0.0001 in the univariate and OR = 1.800, 95% CI 1.349–2.401, *p* = 0.01 in the multivariate regression) but not for bleeding complications (OR = 1.302, 95% CI 0.678–2.499, *p* > 0.05) HTN = predictor for thrombotic events in ET (OR = 1.913, 95% CI 1.099–3.330, *p* = 0.02) and PMF/SMF (OR = 1.960, 95% CI 1.067–3.601, *p* = 0.03)
Kwiatkowski et al. (2021) [13]	Poland	ET	310	ET subjects display a degree of kidney dysfunction even before initiation of risk-adapted treatment (hydroxyurea or anagrelide) the decision to prescribe these drugs is associated with ↑ in creatinine levels ET with concomitant HTN cases had ↑ creatinine concentrations before the commencement of treatment (*p* < 0.001) and experienced ↑ in creatinine levels after cytoreduction with hydroxyurea or anagrelide (0.91 mg/dL pre-treatment versus 0.96 mg/dL post-treatment, *p* < 0.001) ET patients on antihypertensive agents (OR = 2.20, 95% CI: 1.26–3.85, *p* = 0.006) and anagrelide (OR = 13.01, 95% CI: 6.27–27.01, *p* < 0.001), as well as harboring CALR mutations (OR = 1.95, 95% CI: 1.10–3.45, *p* = 0.02), were more likely to experience kidney dysfunction
Person et al. (2021) [37]	Switzerland	PV, ET, PMF	57	post-mortem evaluation of kidneys in MPNs did not delineate proof of HTN or T2DM-related nephropathy in subjects who exhibited diffuse glomerulosclerosis
Lucijanic et al. (2022) [38]	Croatia	PMF, SMF	173	PMF/SMF with high vs. low uric acid levels at diagnosis more likely to suffer from HTN (PMF: ~71% vs. ~52%, *p* = 0.03; SMF: ~65% vs. ~30%, *p* = 0.01)
Akdi et al. (2020) [39]	Turkey	PV	50	PV cases likely to be non-dippers (~64% vs. ~37%, *p* = 0.007) and display lower nocturnal fall in SBP, DBP and MBP (−6.91 ± 8.92 mmHg vs. −11.65 ± 7.68 mmHg, *p* = 0.005; −11.31 ± 12.18 mmHg vs. −16.28 ± 12.03 mmHg, *p* = 0.042; −9.31 ± 10.41 mmHg vs. −14.06 ± 9.41 mmHg, *p* = 0.018) vs. hypertensive non-PV counterparts average-24 h and daytime SBP and DBP similar between the two groups nighttime SBP and DBP elevated in PV (125.32 ± 17.24 mmHg vs. 118.93 ± 12.20 mmHg, *p* = 0.034; 73.74 ± 12.16 mmHg vs. 69.49 ± 8.52 mmHg, *p* = 0.044, respectively) nocturnal falls in SBP (r = 0.305, *p* = 0.002 and *r* = 0.354, *p* < 0.001), DBP (*r* = 0.197, *p* = 0.048 and *r* = 0.236, *p* = 0.017) and MBP (*r* = 0.246, *p* = 0.013 and *r* = 0.293, *p* = 0.003) positively associated with hemoglobin and hematocrit levels
Dobrowolski et al. (2017) [5]	Poland	PV	23	no significant differences in office or 24 h SBP, DBP, heart rate or use of antihypertensive agents in PV vs. healthy counterparts signs of systolic and diastolic dysfunction of the heart by the assessment of several echocardiographic parameters, i.e., septal and lateral systolic velocities (8.7 ± 1.3 vs. 7.2 ± 2.4, *p* = 0.04 and 9.3 ± 1.2 vs. 7.7 ± 2.4, *p* = 0.04, respectively) and global longitudinal, circumferential and radial strains (−20.1 ± 4.3% vs. −18.1 ± 3.1%, *p* = 0.01; −19.7 ± 1.1% vs. −16.7 ± 2.7%, *p* = 0.001; 37.5 ± 8.7% vs. 29.6 ± 12.8%, *p* = 0.05, respectively) elevated in controls, whereas the isovolumic relaxation time was increased in PV (110.9 ± 24.9 ms vs. 83.5 ± 12.9 ms, *p* = 0.0001) hemoglobin levels were negatively associated with the global longitudinal and circumferential strains (β = −0.488, *p* = 0.0001; β = −0.537, *p* = 0.005) in PV hematocrit value was positively linked only with the global longitudinal strain (β = 0.408, *p* = 0.001) in PV red blood cell count was negatively correlated with the isovolumic relaxation time (β = −0.463, *p* = 0.05) in PV
Jóźwik-Plebanek et al. (2020) [40]	Poland	PV	20	HTN affected 75% of PV cases who displayed lower 24 h SBP and 24 h DBP (*p* = 0.003 and *p* = 0.01, respectively) versus comparators All other office or ambulatory BP measurements similar in PV and controls ↓ metanephrine and aldosterone in the plasma (*p* < 0.001 and *p* = 0.008, respectively), ↑ potassium levels (*p* < 0.001), ↓ free normetanephrine, metanephrine and norepinephrine in the urine (*p* = 0.03, *p* = 0.007 and *p* = 0.03, respectively) in PV number of erythrocytes (*p* < 0.001), leukocytes (*p* = 0.001), platelets (*p* < 0.001), hemoglobin (*p* < 0.001) and hematocrit (*p* = 0.02) ↑ in PV ↓ mean corpuscular volume, mean corpuscular hemoglobin concentration and mean corpuscular hemoglobin were lower (*p* < 0.001 for all) PV exhibited reduced capillary blood flow in the retina (*p* = 0.08) which was inversely associated with hemoglobin levels and erythrocyte numbers in PV (r = −0.57; *p* = 0.001 and r = −0.40, *p* = 0.02, respectively aldosterone concentrations negatively correlated with the aforementioned red cell parameters (r = −0.33, *p* = 0.04 for both) although HR baroreflex control was not different in PV and healthy counterparts, daytime, nighttime and 24 h ABPM, in addition to muscle adrenergic nerve activity (*p* = 0.007 for bursts/min and *p* = 0.04 for bursts/100 heartbeats) were ↓ in PV
Rusak et al. (2013) [41]	Poland	PV	73	PV subjects exhibited higher blood viscosity (*p* < 0.01), SBP (*p* < 0.05), DBP (*p* < 0.05), MAP (*p* < 0.05), cell-free hemoglobin (*p* < 0.01) and nitrite/nitrate (*p* < 0.01) versus healthy comparators especially when PV and HTN co-occur cell-free hemoglobin values were positively associated with MAP (*r* = +0.49, *p* < 0.05), nitrite/nitrate (*r* = +0.46, *p* < 0.05), hematocrit (*r* = +0.47, *p* < 0.05) and blood viscosity (*r* = +0.39, *p* < 0.05) isovolemic erythrocytapheresis decreased several biochemical parameters; however, it barely influenced the cell-free hemoglobin—hematocrit, cell-free hemoglobin—blood viscosity associations which continued to display a positive trend line and remained statistically significant (*p* < 0.05)

Legend: NS, not specified. ↑, increase(d). ↓, decrease(d). For abbreviations, see List of abbreviations.

**Table 2 biomedicines-11-00388-t002:** Main results of the investigations linking HTN and thrombosis in MPNs.

Author and Year	Study Location	MPN Subtype	Number of Patients	Main Results
Carobbio et al. (2011) [42]	International cohort: Italy, Austria, Germany, USA	ET	891	HTN, T2DM or smoking (at least one CVRF) were predictors of both major thrombotic events (HR = 1.56, 95% CI = 1.03–2.36, *p* = 0.038) HTN, T2DM or tobacco use were predictors of major arterial thrombotic events (HR = 1.91, 95% CI = 1.19–3.07, *p* = 0.007), namely acute myocardial infarction, ischemic stroke, cerebral transient ischemic attacks or peripheral arterial thrombosis, but not of the occurrence of major venous thrombosis (HR = 0.77, 95% CI = 0.33–1.83, *p* = 0.556), namely venous thromboembolism
Buxhofer-Ausch et al. (2014) [43]	Austria	ET, PMF	167	HTN prevalence similar in both subgroups (50% vs. 44%, *p* = 0.48) HTN = risk factor for thrombosis (univariate model: HR = 3.43, range 1.12–10.52, *p* = 0.03; multivariate model: HR = 3.33, range 0.90–12.29, *p* = 0.07), in particular arterial thrombosis (only in the univariate model: HR = 3.76, range 1.05–13.48, *p* = 0.04; multivariate model: HR = 2.79, range 7.06–11.02, *p* = 0.14) in ET HTN did not impact the occurrence of venous thrombosis (HR = 2.09, range 0.19–23.11, *p* = 0.55) in ET
Pósfai et al. (2015) [44]	Hungary	ET	101	HTN = #1 comorbidity (46.5%) in ET HTN not linked to the occurrence of thrombosis in the logistic regression analysis co-existence of two or more CVRFs out of HTN, dyslipidemia, diabetes or smoking was linked to the development of thrombotic events (*p* = 0.02) thrombosis-free survival lower in ET with ≥ 1 CVRF vs. those without CVRFs (*p* = 0.01) and in ET patients with one CVRF vs. ≥ 2 CVRFs (*p* = 0.002)
Pósfai et al. (2014) [45]	Hungary	ET	128	HTN = predisposing factor (*p* = 0.001) to the development of thrombotic complications in females with ET of whom ~55% (*n* = 70) had elevated BP ≥2 CVRFs = linked with elevated probability of suffering a thrombotic event in women diagnosed with ET (RR = 4.728, 95% CI 1.312–17.040, *p* = 0.01)
Horvat et al. (2018) [46]	Hungary	PV, ET, PMF	258	HTN and presence of ≥1 CVRF = risk factors for thrombotic events (OR = 2.8, 95% CI 1.6–5.0, *p* < 0.001; OR = 3.2, 95% CI 1.7–6.3, *p* = 0.001, respectively), especially arterial thrombosis (OR = 3.3, 95% CI 1.7–6.3, *p* < 0.001; OR = 5.7, 95% CI 2.3–13.9, *p* < 0.001, respectively) in PV (*n* = 70) and PMF (*n* = 54) the presence of ≥1 CVRF but not HTN alone predicted the development of arterial thrombotic complications (OR = 7.9, 95% CI 1.0–64.9, *p* = 0.049; OR = 12.2, 95% CI 0.7–225.3, *p* = 0.044, respectively) both HTN and the presence of ≥1 CVRF were risk factors not only for overall thrombosis (OR = 3.8, 95% CI 1.6–8.7, *p* = 0.003; OR = 5.1, 95% CI 1.8–14.1, *p* = 0.001, respectively), but also for arterial (OR = 2.8, 95% CI 1.2–6.5, *p* = 0.021; OR = 3.9, 95% CI 1.4–11.1, *p* = 0.009, respectively) and venous (OR = 30.3, 95% CI 1.7–532.4, *p* < 0.001; OR = 17.1, 95% CI 1.0–300.8, *p* = 0.005) thrombosis separately in ET
Lekovic et al. (2014) [47]	Serbia	ET	244	~58% of ET cases had HTN development of both arterial and global thrombosis associated with HTN (*p* = 0.01 and *p* = 0.001, respectively), CVRFs in general (*p* = 0.01 and *p* = 0.002, respectively) and number of CVRFs (*p* < 0.001 and *p* < 0.001, respectively)
Lekovic et al. (2015) [48]	Serbia	ET	244	CVRFs (HTN, T2DM and dyslipidemia) and combination of CVRFs and tobacco use were less common in the patients who were still alive at the time of the analysis (~62% versus ~78%, *p* = 0.05 and ~21% versus ~41%, *p* = 0.01, respectively) presence of CVRFs (HR = 2.33) and CVRFs + tobacco use (HR = 2.08) linked with shorter overall survival in ET novel assessment tool for the prognosis of ET, namely the Cardio-IPSET prognostic model which takes into consideration the following factors: age, history of thrombotic events, leukocyte count and the presence of CVRFs (HTN, T2DM, dyslipidemia, and smoking) ~75% of deaths in ET attributed to cardiovascular causes
Schwarz et al. (2015) [49]	Czech Republic	PV, ET, PMF	1179	HTN = predictor of overall thrombosis (*p* = 0.003), major thrombosis (*p* = 0.022) and arterial thrombosis (*p* < 0.001); however, not of microvascular events or venous thrombotic events based on the univariate analysis in MPNs treated with anagrelide in the multivariate regression analysis, HTN was the best predictor of arterial thrombotic events (OR = 1.813, 95% CI 1.295–2.538, *p* = 0.001)
Accurso et al. (2020) [50]	Italy	PV, ET	403	HTN = #1 cardiovascular comorbidity in PV and ET (~64%) an elevated percentage of PV vs. ET cases (~39% vs. ~27%, *p* = 0.014) experienced thrombotic complications CVRFs associated with decreased survival in PV (*p* = 0.014) and ET (*p* = 0.036)
Cucuianu et al. (2006) [51]	Romania	PV, ET	37	~31% of the patients had HTN association of HTN, platelet count > 600,000 platelets/mmc and hematocrit > 55% was linked with higher incidence of thrombotic events (*p* = 0.02) in PV
Barbui et al. (2017) [52]	International cohort: Italy, Austria, USA	PV	604	HTN impacts the incidence of thrombosis in low-risk PV (*n* = 525). Thrombosis-free survival higher in low-risk PV patients who did not suffer from HTN (IR = 0.85, 95% CI 0.57-1.25 vs. IR = 2.05, 95% CI 1.34-3.14, *p* = 0.025) Compared to another ET cohort (*n* = 891), HTN was more prevalent in PV (OR = 1.38, *p* = 0.022) and BP values positively correlated with hematocrit levels
Benevolo et al. (2021) [53]	Italy	PV	861	HTN (HR = 1.77, 95% CI 1.03–3.06, *p* = 0.04) and previous history of thrombosis (HR = 2.10, 95% CI 1.21–3.60, *p* = 0.01) elevate risk of thrombosis in PV
Birgegård et al. (2018) [54]	International cohort: Sweden, Italy, France, UK, USA, Germany, Spain, Switzerland	ET	3649	post-hoc multivariate analysis of the Evaluation of Anagrelide Efficacy and Long-term Safety study, long-term research with prospective observational design which recruited high-risk ET cases 34% of ET cases had elevated BP (#1 CVRF in ET) HTN = predictor of major hemorrhages (HR = 1.33, 95% CI 1.04–1.69, *p* = 0.02) and thrombohemorrhagic complications (HR = 1.69, 95% CI 1.02-2.79, *p* = 0.04)
Cerquozzi et al. (2017) [55]	USA	PV	587	42% of PV cases had HTN rate of arterial and venous thrombotic complications was elevated in subjects with elevated BP (52% vs. 38%, *p* = 0.004 and 44% vs. 30%, *p* = 0.009, respectively). Individuals with PV had lower thrombosis-free survival (HR = 1.7, 95% CI 1.1–2.6, *p* = 0.02) in the univariate but not in the multivariate analysis
Cervantes et al. (2006) [56]	Spain	PMF, SMF	155	patients with any CVRF (HTN, T2DM, hypercholesterolemia, use of cigarettes) were at an elevated risk for thrombosis (OR = 14.9, 95% CI 2.5–87, *p* = 0.003) and had lower thrombosis-free survival (~83% vs. 97%, *p* = 0.02)
Navarro et al. (2015) [57]	Brazil	ET	46	association between CVRFs and thrombosis (*p* = 0.01), namely arterial (*p* = 0.03) and not venous (*p* > 0.05) thrombotic complications
Shih et al. (2002) [58]	Taiwan	ET	89	assessment of thrombosis in women with ET and with/without clonal/polyclonal X-chromosome inactivation patterns Thrombosis but not hemorrhage was more common in ET subjects with vs. without HTN (*p* = 0.002 and *p* = 0.287, respectively) After adjustment for HTN and age, the risk of thrombotic events was 7 times more elevated in ET individuals with clonal X-chromosome inactivation patterns vs. those without
Bucalossi et al. (1996) [59]	Italy	PV, ET	81	similar prevalence of HTN in PV and ET with/without thrombosis
Landolfi et al. (2007) [60]	International cohort	PV	1638	assessment of 1638 subjects from the European Collaboration on Low-Dose Aspirin in Polycythemia Vera (ECLAP) HTN did not emerge as a predictor for major/arterial/venous thrombosis, AMI, TIA, stroke or peripheral arterial thrombosis
Finazzi (2004) [61]	International cohort	PV	1630	European Collaboration on Low-Dose Aspirin in Polycythemia Vera (ECLAP) analysis cumulative incidence rate of cardiovascular events (i.e., cardiovascular death and non-fatal thrombotic events) = 5.5 events/100 persons per year Thrombosis = main cause of death Age > 65 years, history of thrombosis = predictors of cardiovascular events smoking, HTN, congestive heart failure = risk factors for thrombosis Platelet counts, myelosuppressive drugs = no association with the risk of cardiovascular events Antiplatelet treatment = only variable associated with lower risk of thrombosis
Bazzan et al. (1999) [62] Cortelazzo et al. (1990) [63]	Italy	ET	187 100	HTN did not impact thrombosis-free survival and life expectancy
Jantunen et al. (2001) [64]	Finland	ET	132	cigarette use = more common risk factor for thrombosis versus HTN (24.3% versus 20.5%) male gender (*p* < 0.001) and tobacco consumption (*p* = 0.01) = risk factors for thrombotic complications, whereas HTN did not (*p* = 0.34)
Barbui et al. (2018) [65]	International cohort	PV, ET, PMF	597	HTN = more common occurrence in MPNs who developed ischemic stroke versus those with transient ischemic attacks HTN = prognostic factor in recurrence of stroke (HR = 4.24)
Košťál et al. (2020) [66]	Czech Republic	PV, ET, PMF	1442	HTN = more common individuals who experienced a stroke or a TIA (~53% vs. ~41%), HTN = risk factor for such complications based on the univariate analysis model (OR = 1.604, 95% CI = 1.219–2.111, *p* = 0.001) but not on the multivariate logistic models on data with imputed missing values (OR = 1.170, 95% CI 0.845–1.619, *p* = 0.344 for treated and untreated subjects; OR = 0.918, 95% CI = 0.55–1.534, *p* = 0.745 for subjects not receiving cytoreductive agents
De Stefano et al. (2018) [67]	International cohort	PV, ET, PMF	597	assessment of MPNs with history of stroke or TIA similar HTN frequency (stroke vs. TIA = 57% vs. 52%, *p* > 0.05) HTN = independent risk factor for the recurrence of ischemic stroke in MPNs (HR = 4.24, 95% CI 1.23–14.7) Cytoreduction decreased the risk of stroke re-occurrence by 76%
Jiao et al. (2021) [68]	China	ET	91	HTN more prevalent (~32% vs. ~4%, *p* = 0.003) in ET without CVST
Robertson et al. (2007) [69]	UK	PV, ET, PMF	118	compared to subjects with HTN, individuals diagnosed with MPNs display elevated concentrations of soluble *p*-selectin (*p* < 0.001), particularly if they harbor the *JAK2V617F* mutation (*p* = 0.006 between *JAK2V617F*-positive and *JAK2V617F*-negative cases), and D-dimers (*p* = 0.03), but similar soluble E-selectin, thrombin–antithrombin complexes, prothrombin fragments or antiphospholipid antibodies soluble *p*-selectin levels were similar in MPN patients who experienced thrombotic events versus those who did not

Legend: NS, not specified. For abbreviations, see list of abbreviations.

## Data Availability

Not applicable.

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
