# Peer review of "Primary Arterial Hypertension and Drug-Induced Hypertension in Philadelphia-Negative Classical Myeloproliferative Neoplasms: A Systematic Review"

_biomedicines, 2023, doi:10.3390/biomedicines11020388_

Round 1

Reviewer 1 Report

The paper named” Primary Arterial Hypertension and Drug-Induced Hypertension in Philadelphia-Negative Classical Myeloproliferative Neoplasms: A Systematic Review” is a well done review about the presence of Hypertension in patients with blood diseases. The paper is easy to read, and provide lot of information. Moreover it is well divided in sections that allow a better compression.

Only minor questions are required.

1)      Tthroughout the text there are many abbreviations that it would be necessary to indicate its long name the first time it is cited.

2)      The references are confusing, because sometimes the author name and the date are cited and other times only the author name. Moreover in the reference section the papers are numerated. Please clarify this point

3)      A summary table showing the more concise information is required in order to see quickly the decider information.

4)      It will be interesting if at the end of each section a brief conclusion is shown

Author Response

Dear Academic Editor,

Dear Reviewer 1,

We are very thankful to you for the pertinent note. We have carefully read your comments and have revised/completed the manuscript accordingly. Our responses are given in a point-by-point manner below. All the changes to the manuscript are highlighted in yellow.

We hope that, in this new form, the manuscript will be suitable for publication in Biomedicines.

All in all, we thank you for your constructive comments regarding our manuscript.

Reviewer 1

The paper named” Primary Arterial Hypertension and Drug-Induced Hypertension in Philadelphia-Negative Classical Myeloproliferative Neoplasms: A Systematic Review” is a well done review about the presence of Hypertension in patients with blood diseases. The paper is easy to read, and provide lot of information. Moreover it is well divided in sections that allow a better compression.

Only minor questions are required.

Response: We thank you for your constructive comments regarding our manuscript.

1)      Tthroughout the text there are many abbreviations that it would be necessary to indicate its long name the first time it is cited.

Response: Thank you for pointing this out. We have ensured that all abbreviations are explained when used for the first time. In addition, a list of abbreviations has been added at the end of the paper.

2)      The references are confusing, because sometimes the author name and the date are cited and other times only the author name. Moreover in the reference section the papers are numerated. Please clarify this point

Response: Thank you for pointing this out. This was an error of the software used to format the references. We have arranged all references according to the requirements of the journal.

 3)      A summary table showing the more concise information is required in order to see quickly the decider information.

Response: Thank you for this excellent suggestion. We have added summary tables for the main sections.

4)      It will be interesting if at the end of each section a brief conclusion is shown

Response: Thank you for this excellent suggestion. A brief conclusion was added to the Discussions section.

Reviewer 2 Report

The authors uncovered the role of primary arterial hypertension and drug-induced hypertension in philadelphia-negative classical myeloproliferative neoplasm.

The manuscript is of interest.

Points to be addressed:

1) The rationale of why the authors came up with this research is scanty and is related to a lack of novelty: please highlight what this manuscript might add.

2) What is the information that is not exactly available that motivated the authors to come up with this information. What are the current caveats and how do the authors highlight the current research in answering them? If not they need to address in background and infuture directions .

3)State of the art figures are required: scale bar should be provided in high resolution.

4)The authors could provide a little more consideration of genomic directed stratifications in clinical trial design and enrolments. 

5)The underlying message here is that more precision and individualized approaches need to be tested in well-designed clinical trials – a challenge, but I would be interested in their perspective of how this might be done. If beyond the scope of the manuscript, this should be highlighted as a limitation

6)Level of consistency among studies should be better reported; please provide

7)The authors need to highlight what new information the review is providing to enhance the research in progress: in introduction  and discussion sections this reviewer personally misses few insights regarding the role of microenvironment in mediating key biological checkpoint of MPN:   Inflammation and its effects in the bone marrow microenvironment represent a paradigmatic condition in which the hematopoietic niche and the immune systems, thought to properly sustain blood cell production and distinguish between friend and foe, can actively sustain a corrupted neighborhood within a chronic aberrant inflamed state. The bone marrow niche hijacks the physiologic hematopoiesis. The interactions between the hematopoietic stem cells and the niche in the bone marrow are critical determinants of quiescence.  Philadelphia negative myeloproliferative neoplasms, and chronic myeloid leukemia share aging with variable fitness and hematopoietic stem cell attrition, extrinsic stress, enhanced stressor-specific fitness, and intrinsic defect across the hematopoietic process represent the route for novel insights in defective hematopoiesis and potentially hypertension (please refer to 10.20517/2394-4722.2021.166  and expand). This can boost the discussion and envision potential novel therapeutic windows 

Author Response

Dear Academic Editor,

Dear Reviewer 2,

We are very thankful to you for the pertinent note. We have carefully read your comments and have revised/completed the manuscript accordingly. Our responses are given in a point-by-point manner below. All the changes to the manuscript are highlighted in yellow.

We hope that, in this new form, the manuscript will be suitable for publication in Biomedicines.

All in all, we thank you for your constructive comments regarding our manuscript.

Reviewer 2

The authors uncovered the role of primary arterial hypertension and drug-induced hypertension in philadelphia-negative classical myeloproliferative neoplasm.

The manuscript is of interest.

Points to be addressed:

1) The rationale of why the authors came up with this research is scanty and is related to a lack of novelty: please highlight what this manuscript might add.

2) What is the information that is not exactly available that motivated the authors to come up with this information. What are the current caveats and how do the authors highlight the current research in answering them? If not they need to address in background and in future directions.

Response: Thank you for this excellent suggestion. We have highlighted the novelty added by this manuscript, as well as highlighted the current caveats and the role of the current research in answering some of them.

Considering that MPNs are generally blood cancers encountered in the elderly population who also displays numerous cardiovascular risk factors (CVRF), e.g., HTN, type 2 diabetes mellitus (T2DM), dyslipidemia (in particular hypercholesterolemia), smoking and others, and subsequent cardiovascular disease, understanding the epidemiology and the impact of each CVRF on the risk of thrombosis and on survival requires further investigation [14,15]. A current caveat of the research focused on the assessment of CVRF in MPNs is that most investigations published so far have explored the contribution on thrombotic events and risk of death of CVRF as a singular entity, although their epidemiology in MPNs is probably distinct, similar at least to the prevalence of CVRF in the adult population. As HTN is the most common CVRF encountered in the general adult population [16], we believe that analyzing its epidemiology, as well as contribution to thrombosis occurrence and impact on survival, should emerge as the first step in tracking down the relevance of each CVRF in MPNs. In addition, physicians should also be aware of drug-induced HTN which can develop as a side effect of the medications currently employed to treat individuals suffering from MPNs. Thus, the impact of primary arterial HTN in MPNs, however, still remains unclear, with scanty literature available, mostly focusing on CVRF as a singular entity or on organ-specific HTN. Furthermore, available studies reporting findings on drug-induced HTN in MPNs have also depicted varying and contradictory findings. In consideration of the above, this study set out to systematically review available literature and shed light on the occurrence of HTN in MPN and its association with thrombosis, as well as the impact on BP of the drugs used in MPN management. To our view, one of the main novelties of this manuscript is that it is the first systematic review to provide a comprehensive overview of primary arterial HTN and drug-induced HTN in MPNs.

3) State of the art figures are required: scale bar should be provided in high resolution.

Response: Thank you for this excellent suggestion. We have created a state-of-the-art figure for this manuscript – see Figure 2. We did not use scale bars in this paper.

Figure 2: The pathogenesis of myeloproliferative neoplasms is multifaceted and often includes genetic mutations involving the JAK2, CALR, and MPL genes among others. Abnormal clone–derived blood cells (i.e., erythrocytes, platelets, and leukocytes) drive prothrombotic features. The secretion of pro-inflammatory cytokines and reactive-oxygen species disrupts normal endothelial function and leads to increased expression of endothelial adhesion receptors. The release of tissue factor from activated platelets aids in the formation of fibrin clots and DNA released from activated neutrophils forms neutrophil-extracellular traps (NETs). Additionally, red blood cell (RBC) aggregation occurs through biochemical changes in the RBC membrane. The culmination of these maladaptive events causes increased blood viscosity, vascular resistance, and arterial hypertension through Poiseuille's law.

4) The authors could provide a little more consideration of genomic directed stratifications in clinical trial design and enrolments. 

5) The underlying message here is that more precision and individualized approaches need to be tested in well-designed clinical trials – a challenge, but I would be interested in their perspective of how this might be done. If beyond the scope of the manuscript, this should be highlighted as a limitation.

7) The authors need to highlight what new information the review is providing to enhance the research in progress: in introduction  and discussion sections this reviewer personally misses few insights regarding the role of microenvironment in mediating key biological checkpoint of MPN:   Inflammation and its effects in the bone marrow microenvironment represent a paradigmatic condition in which the hematopoietic niche and the immune systems, thought to properly sustain blood cell production and distinguish between friend and foe, can actively sustain a corrupted neighborhood within a chronic aberrant inflamed state. The bone marrow niche hijacks the physiologic hematopoiesis. The interactions between the hematopoietic stem cells and the niche in the bone marrow are critical determinants of quiescence.  Philadelphia negative myeloproliferative neoplasms, and chronic myeloid leukemia share aging with variable fitness and hematopoietic stem cell attrition, extrinsic stress, enhanced stressor-specific fitness, and intrinsic defect across the hematopoietic process represent the route for novel insights in defective hematopoiesis and potentially hypertension (please refer to 10.20517/2394-4722.2021.166  and expand). This can boost the discussion and envision potential novel therapeutic windows.

Response: Thank you for this excellent suggestion. We have added these points to the discussion and referenced the paper kindly provided by you.

4.5 Caveats and Future Research Directions in Unravelling the Interplay between Hypertension and Myeloproliferative Neoplasms

 To our knowledge, the present manuscript is the first systematic review to examine the interplay between primary arterial HTN and drug-induced HTN in MPNs. Based on our findings, we can conclude that HTN is the most common comorbidity in the aforementioned blood cancers, influencing not only the risk of thrombosis and the management strategies, but also the survival of patients who are diagnosed with both these disorders.

However, future research should also focus on not only on understanding the clinical relevance of the association between MPNs and HTN, but also on molecular assessments exploring their interplay. For example, genomic-directed stratification, preferably in the setting of randomized controlled trials design and enrolment, but also of cohort investigations, might aid us in deciphering the complex crosstalk between CVRF, cardiovascular disease and MPNs. In these hematological malignancies, cardiovascular disease has mainly been linked to the presence of the JAK2V617F mutation which is the most frequently encountered genetic alteration in MPNs. Thus, a potential objective of upcoming investigations should be based on genomic-driven stratifications in order to depict the relationship of other genetic changes in MPNs with HTN. For example, little is known today regarding the contribution to cardiovascular disease onset other somatic mutations recognized in MPNs subjects, i.e., mutations in the calreticulin (CALR) or myeloproliferative leukemia protein (MPL) driver genes, or about the interplay between HTN and genetic alterations in epigenetic regulators, e.g., DNA methyltransferase 3 alpha (DNMT3A), tet methylcytosine dioxygenase 2 (TET2), additional sex combs like transcriptional regulator 1 (ASXL1), isocitrate dehydrogenase 1 or 2 (IDH1/2) genes [14,15,99,100]. The use of genomic stratification is of paramount importance in hematology and oncology, where the application of precision medicine, and individualized approaches can aid better prognostication and treatment selection, including targeted therapies and candidate selection for bone marrow transplantation. For example, patients with MPNs can benefit from the use of next-generation deoxyribonucleic acid (DNA) sequencing-based gene panels or liquid biopsy-based tools that are able not only to stratify patients based on genetic alterations, but also predict the risk of thrombosis, progression to secondary myelofibrosis (PV-MF or ET-MF), transformation into acute myeloid leukemia (AML), and death [99,101,102,103]. However, a main limitation of MPN management worldwide is that genomics, proteomics and other sophisticated laboratory studies are not available everywhere in the world, largely due to the elevated costs and requirement for trained personnel to implement and handle such experiments. Therefore, clinical judgment combined with morphology (cytology and most importantly histopathology) studies and genetic testing for JAK2, CALR and MPL mutations remain mandatory in the accurate diagnosis of MPNs globally [104]. Hence, assessment and correct targeting of CVRF, and in particular of HTN, are a must in the management of MPNs, and we believe that our systematic review has provided insightful information regarding the impact of this comorbidity in the aforementioned myeloid malignancies.

The pathogenesis of MPNs is complex and their onset requires the contribution of an intricate network of molecular mechanisms, as highlighted in Figure 2. Interestingly, Solimando et al. have hypothesized that clonal hematopoiesis of indeterminate potential (CHIP) can contribute to the development of chronic inflammation that hijacks the bone marrow microenvironment ecosystem and can eventually elevate the risk of MPN development [105]. The interplay between CHIP and inflammation has been demonstrated to lead not only to the development of hematological malignancies, e.g., MPNs, myelodysplastic syndromes, de novo or secondary AML, but also to systemic effects and the onset of cardiometabolic disorders, e.g., HTN, T2DM, cardiovascular disease etc. [15,100,105]. Thromboinflammation and its crosstalk with somatic mutations (JAK2V617F in particular), epigenetic regulators, splice factor modifiers and oxidative stress are well-established molecular events that drive the occurrence of thrombotic events, as well as phenotypic transformation to PV-MF or ET-MF and eventually accelerated/blastic phase MPNs. Consequently, future approaches to therapeutic targeting in MPNs and acute leukemia should also concentrate on and exploit the antioxidant and anti-inflammatory properties of current MPN medication, e.g., ropeginterferon alpha-2b or JAK1/2 inhibition, as well as potentially synthesize novel compounds from natural products that exhibit these properties when available [106,107,108,109,110,111]. Moreover, upcoming investigations should also aim to dissect the relationship between CHIP and MPNs in order to identify CHIP individuals at risk of MPN development, and in particular those at higher risk to eventually be diagnosed with secondary AML [112,133].

Our paper have several strengths and limitations. Firstly, to our knowledge, it is the first systematic review to examine the relationship between MPNs and primary arterial HTN, as well as drug-induced HTN. As the assessment was based on data extracted from publications available in PubMed/MEDLINE, Web of Science and SCOPUS, we believe the presented data is robust. Main limitations of our research is the impossibility to compute a meta-analysis due to the heterogeneity of the data, as most of the studies were observational in design, and the fact that HTN was not necessarily a primary outcome of the investigated manuscripts. However, we are confident that this systematic review sheds more light into the association of HTN and MPNs, arguing for a better CVRF assessment and management in subjects suffering from MPNs.      

6) Level of consistency among studies should be better reported; please provide

Response: Thank you for this suggestion, however, it is beyond the scope of our paper which was mainly based on observational studies and not RCTs. Thus, it is difficult to perform this analysis which, even if reported, would not add much to the content of our manuscript. HTN was not an outcome of these papers, we gathered information related to the interplay of HTN and MPNs based on the content of the eligible publications. We have highlighted this as a limitation.

Round 2

Reviewer 2 Report

The authors have clarified several questions I raised in my previous review. Most of the major problems have been addressed by this revision.